# Transforming Language Models into Program Interpreters via Execution Trace Chain of Thought

**Koshi Eguchi**[1,†]                                                 *koshi@sakana.ai*
**Kazusato Oko**[1,2,†]                                               *oko@berkeley.edu*
**Kenshin Yamauchi**[1]                                  *kenshin121026@gmail.com*
**Makoto Shing**[1]                                                 *mkshing@sakana.ai*
**Takuya Akiba**[1]                                                   *takiba@sakana.ai*
[1] *Sakana AI,* [2] *University of California, Berkeley,* [†] *Equal contribution*

**Reviewed on OpenReview:** *https://openreview.net/forum?id=pOg7iub4Pz*

## Abstract

Code execution reasoning (CER), the ability to predict how code executes on a given input, has been added to the expected aspects of language models' coding capabilities. However, many open-source models perform poorly on simple code snippets and, as our observations show, they exhibit limitations even on a single basic operation. To enable language models to accumulate fine-grained reasoning steps in a structured format, we propose leveraging extremely granular execution traces as chain-of-thought rationales. Specifically, we introduce a fine-tuning method called *ET-CoT* (Execution Trace Chain of Thought), which leverages execution traces generated by our custom code interpreter and characterized by sub-line-level, thorough expansion of all expressions, going beyond merely logging intermediate variables. After fine-tuning with 127k examples, ET-CoT effectively improves CER performance, for instance with Qwen2.5-7B-Instruct outperforming its official Coder model. In addition, our custom tests show improved accuracy on repeated application of simple operations. Overall, ET-CoT serves as a unique approach that provides valuable insight into how systematically composing atomic reasoning steps improves CER performance.

## 1 Introduction

With growing expectations for language models (LMs) to carry out the full cycle of code generation, debugging, and optimization with minimal human intervention (Islam et al., 2024; Novikov et al., 2025), the coding-related capabilities required of them now extend beyond code generation alone (Hou et al., 2024). Code execution reasoning (CER) (Austin et al., 2021; Nye et al., 2021), the ability to simulate how a code snippet executes on given inputs, is now regarded as one such capability. Such understanding of the concrete behavior of code is a natural competence of skilled human programmers, and it is critical for debugging and repairing errors in generated code (Gu et al., 2024a). Therefore, CER benchmarks have been included in evaluation suites for state-of-the-art general-purpose (Yang et al., 2025) and code-specific (Hui et al., 2024) LMs.

However, CER remains difficult for many open-source models (Ding et al., 2024a), and even frontier models can struggle on relatively simple problems (Gu et al., 2024b). Sources of CER's difficulty and effective ways to address them remain unclear (La Malfa et al., 2024). This lack of clarity arises partly because the training data is often undisclosed even for open-weight models (Groeneveld et al., 2024), making it hard to assess how datasets influence CER performance, and partly because prior work has mostly treated CER as just one objective among many (Lozhkov et al., 2024; Li et al., 2025b; Ma et al., 2025), so its improvement has rarely been investigated in isolation. One commonly attempted approach is to fine-tune LMs on natural-language explanations of code execution (Ni et al., 2024; Ding et al., 2024a; Li et al., 2025b). Nevertheless, even this approach is not yet well established, as Wang et al. (2025) report that it yielded only small gains over direct-output fine-tuning, casting doubt on whether true semantic understanding is being acquired.

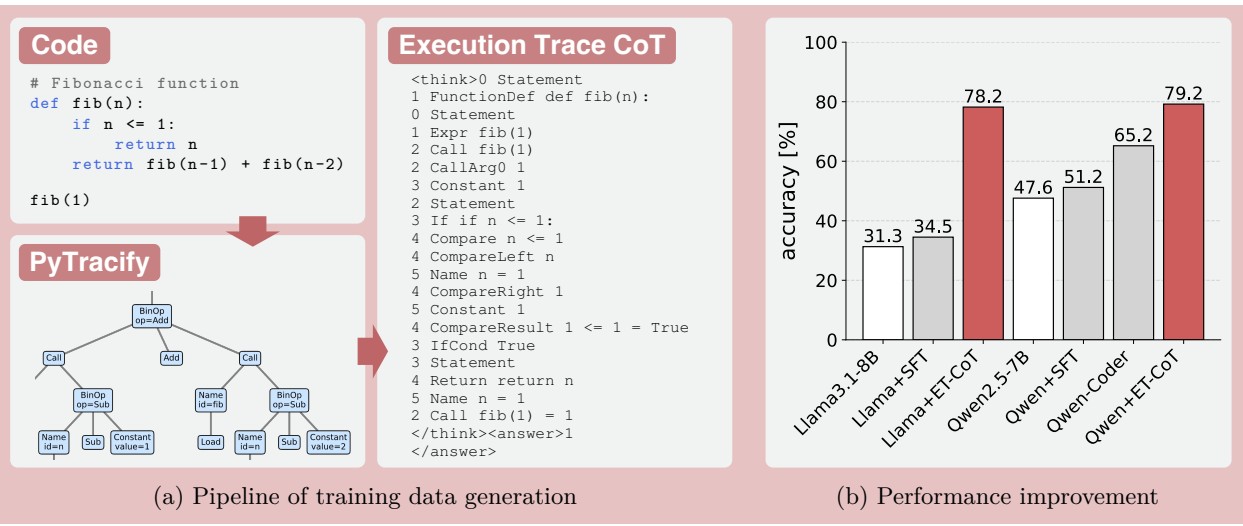

Figure 1: **Execution Trace Chain of Thought (ET-CoT).** (a) We convert Python code snippets into extremely detailed execution traces by our custom interpreter, PyTracify, and use these traces as chain-of-thought rationales for fine-tuning. (b) We report the average accuracy of CRUXEval and LiveCodeBench (execution). It shows that ET-CoT effectively turns 8B-class LMs into strong code interpreters.

In this context, this paper studies CER in isolation to better understand its bottlenecks and to obtain actionable insights for improving performance. We begin by showing that CER tasks can be more challenging for LMs than explaining code functionality in natural language, and that even predicting the result of a single basic operation may require multi-step chain-of-thought (CoT) reasoning. Based on these observations, we hypothesize that decomposing reasoning to the sub-line level and teaching the semantics and systematic composition of minimal execution steps can improve CER performance. To examine this hypothesis, we introduce ET-CoT (Execution Trace Chain of Thought), a fine-tuning method that utilizes execution traces that thoroughly expand every operation and are produced by a custom full-fledged interpreter. Deterministic reasoning via execution traces is unconventional but aligns with the deterministic nature of the task and enables isolated evaluation of the hypothesis by controlling reasoning granularity. The results demonstrate the effectiveness of ET-CoT, easily converting models that previously lacked even basic syntactic understanding into strong code interpreters (Figure 1 (b)).

The contributions of this work are summarized as follows:

- We begin by testing LMs' understanding of basic operations, evaluating both their CER performance and their accuracy in explaining code in natural language such that its functionality can be reproduced (Section 3). We observe that CER is often more challenging than natural language explanations and requires long multi-step CoT reasoning despite the apparent simplicity of the problems. This tendency is particularly pronounced in non-reasoning models of 8B parameters or smaller.

- We develop a pipeline for generating synthetic CoT rationales of detailed execution traces. We first curate 127k unique Python code snippets, by combining collected samples with additional snippets that we generate ourselves (Sections 4.1, 4.2). We then develop our custom Python interpreter *PyTracify* to generate execution traces. Notably, our trace format thoroughly decomposes execution at a sub-line level of granularity, not only logging variable updates but also performing detailed evaluations of individual expressions such as comparisons and condition checks (Section 4.3). We refer to fine-tuning with the execution-trace data obtained as ET-CoT (Execution Trace Chain of Thought).

- We fine-tune various LMs ranging from 0.5B to 8B parameters and observe that ET-CoT effectively transforms LMs that originally lacked an understanding of code execution into strong code interpreters (Section 5.1). For example, ET-CoT raises Qwen2.5-7B-Instruct (Qwen team, 2025) to 70.0% on CRUXEval-O (Gu et al., 2024b) and 88.3% on LiveCodeBench (execution) (Jain et al., 2025), surpassing the official code-oriented variant, Qwen2.5-Coder-7B-Instruct (Hui et al., 2024). Our ablation studies show that the thorough breakdown of PyTracify is key to these gains (Section 5.4).

- We also quantitatively evaluate the ability to consistently repeat simple procedures by designing a test whose iteration complexity is configurable (Section 5.3). While base models exhibit pronounced errors especially in the initial steps, models trained with ET-CoT mitigate this instability.

## 2 Related works

**Benchmarks and challenges of CER.** Apart from general-purpose coding benchmarks such as MBPP (Austin et al., 2021), datasets dedicated to CER have recently appeared, reflecting the growing attention to CER. LiveCodeBench (execution) (Jain et al., 2025) collects competitive programming problems, while CRUXEval (output prediction) (Gu et al., 2024b) is a synthetic dataset generated with Code Llama and recently extended to multiple languages (Xu et al., 2025). Both contain code snippets of about ten lines with input–output pairs. These datasets reveal surprising weaknesses of LMs in CER, and prior work has identified difficulties in handling snippets with multiple operators and control flow, as well as error accumulation as the critical path grows (Chen et al., 2025; La Malfa et al., 2024; Liu et al., 2025; Liu & Jabbarvand, 2025). However, those discussions mainly consider problems where even commercial models fail (though sometimes presented as simple (Gu et al., 2024b)), and the analyses of 8B-class models have often been limited to observing that they generally fail on these problems. It has also been reported that CER performance is only weakly correlated with code generation ability (Austin et al., 2021; Gu et al., 2024a; Luo et al., 2024; Wei et al., 2024). Consequently, recent technical reports evaluate CER performance alongside generation metrics (Hui et al., 2024; Yang et al., 2025), highlighting the importance of CER as an independent task.

**Fine-tuning to improve the CER ability.** Fine-tuning with direct code, input, and output pairs was attempted in Austin et al. (2021); Gu et al. (2024b), yielding only limited improvements. To further improve CER capability, recent work has attempted to incorporate intermediate reasoning steps. In natural language, NExT (Ni et al., 2024) fine-tuned LMs on execution-aware rationales, while SemCoder (Ding et al., 2024a) enhanced the semantic understanding of code with step-wise explanations. Li et al. (2025b) scaled this approach by developing an automatic pipeline to generate such CoT data. On the other hand, systematic logging of intermediate variables was utilized by Scratchpad (Nye et al., 2021) (but they used the same dataset for both training and evaluation). Similar logging formats were also used by CodeExecutor (Liu et al., 2023) and Ding et al. (2024b) in pretraining. However, Wang et al. (2025) reproduced CodeExecutor, NExT, and SemCoder and reported that they were not more effective than SFT without traces in their setting.

Comparing ET-CoT with these approaches, natural language explanations exhibit variability, whereas ET-CoT enables models to build up reasoning in a more consistent, structured manner. Also, per-line logs of variables do not align with the observation that multi-step reasoning is required even for a single operation, whereas ET-CoT decomposes execution at the sub-line, operation level, addressing this limitation.

**Theoretical backgrounds.** Theoretical results also motivate approaches to code reasoning with CoT, particularly those that mechanically simulate the underlying computational steps. Transformers are Turing complete under suitable modifications (Pérez et al., 2019; Pérez et al., 2021), and increasing the length of CoT allows them to solve progressively broader classes of problems (Xu & Sato, 2025; Schuurmans et al., 2024; Bhattamishra et al., 2020). Concretely, problems solvable by a Turing machine within a given time bound can be solved using a chain of thought whose length is of the same order (Merrill & Sabharwal, 2024). On the other hand, without CoT of sufficient length, the expressivity of Transformers collapses to a relatively simple class of problems (low-level circuit classes) (Merrill & Sabharwal, 2023). In Appendix E, we further discuss examples of problems that inherently require long CoT, supporting the use of detailed execution traces in ET-CoT. We also remark that Zhai et al. (2024) proved that Transformers can efficiently process compiler tasks such as AST construction, symbol resolution, and type analysis.

## 3 Limitations of code execution reasoning even at the level of basic operations

We first take a closer look at cases where language models fail at code execution reasoning. Most existing datasets for CER are designed to challenge advanced commercial LLMs by combining multiple operations (Ma et al., 2023; Gu et al., 2024b; Jain et al., 2025). However, such complications may obscure more fundamental

difficulties in code execution reasoning. Therefore, we begin by introducing a dedicated test to expose the limitations of CER at the level of single-operation understanding, the minimal unit of code execution reasoning.

Based on the CRUXEval (Gu et al., 2024b) dataset, we generated 800 test problems. Specifically, from each original snippet of up to 13 lines, we selected the most essential operation, defined it as a function, and prepared corresponding inputs and outputs. An example is shown in Figure 2 (a case mispredicted by GPT-4.1), with pipeline details provided in Appendix B.1.

Function:

```
def f(o):
    return o[-2::-1]
```

Input:

```
o = 'bab'
```

Figure 2: Sample problem.

We tested 21 models on this dataset (Figure 3). Red denotes predictions with answer-only output, while blue denotes predictions with CoT reasoning. Green indicates the correctness of natural language function descriptions. Concretely, each model was asked to describe the function's operation in natural language, and GPT-4.1 then applied this description to the inputs and checked whether the output matched. Refer to Appendix B.2 for more details.

We briefly summarize the observations as follows: **(i) Multi-step CoT is crucial even for simulating a single operation**: for many models, CoT improves accuracy by more than 10% over direct prediction. **(ii) Understanding of basic operations remains challenging across non-reasoning models**: in non-reasoning models from 1B to 8B params, accuracy remains below 75% even when CoT is applied to a single operation. In contrast, the open reasoning models achieve performance comparable to commercial models such as Gemini 2.0 and GPT-4.1. **(iii) The ability to provide natural-language descriptions does not imply procedural understanding**: models can provide accurate natural language descriptions of operations, sufficient to derive the correct output, but their ability to actually apply these operations to inputs diverges substantially from their accuracy in describing them in natural language.

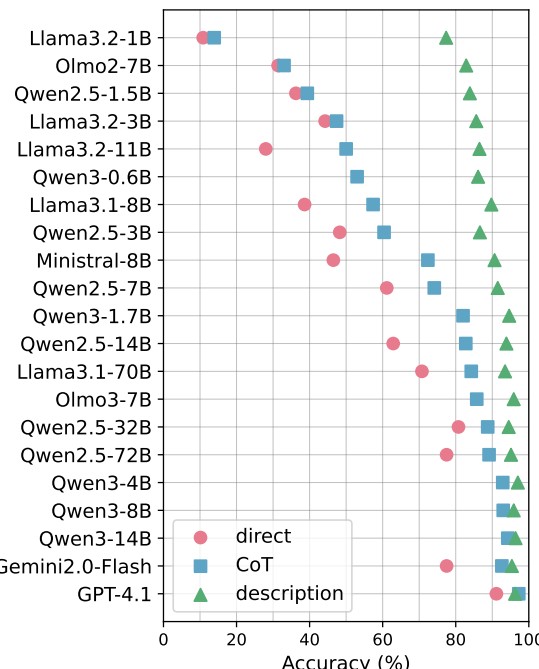

Figure 3: Accuracy of execution simulation for single-operation functions.

Given these observations, we hypothesize that code execution reasoning ability can be improved by decomposing the reasoning process into a multi-step CoT composed of sub-line-level steps, teaching models the semantics of minimal execution steps in basic operations, and enabling their systematic accumulation. However, they are unlikely to be resolved by natural language code explanations or by simply logging per-line updates of intermediate variables. Natural language explanations are inconsistent in granularity of reasoning and phrasing, and only weakly connected to procedural understanding as we found above. Per-line logging of variables still lacks sufficient resolution, as even a single operation can require multi-step reasoning.

Therefore, we came to consider using execution traces as a systematic way to consistently accumulate sub-line level reasoning steps. In the following, we design execution traces with our custom code interpreter, and train models on them. We then investigate how effectively this approach improves CER performance, particularly in small non-reasoning models where the above limitations are most pronounced, to test our hypothesis.

## 4 Fine-tuning with Execution Trace Chain of Thought

We propose ET-CoT (Execution Trace Chain of Thought), a fine-tuning framework that leverages detailed execution traces as CoT rationales. To generate traces, we designed PyTracify, a full-fledged code interpreter

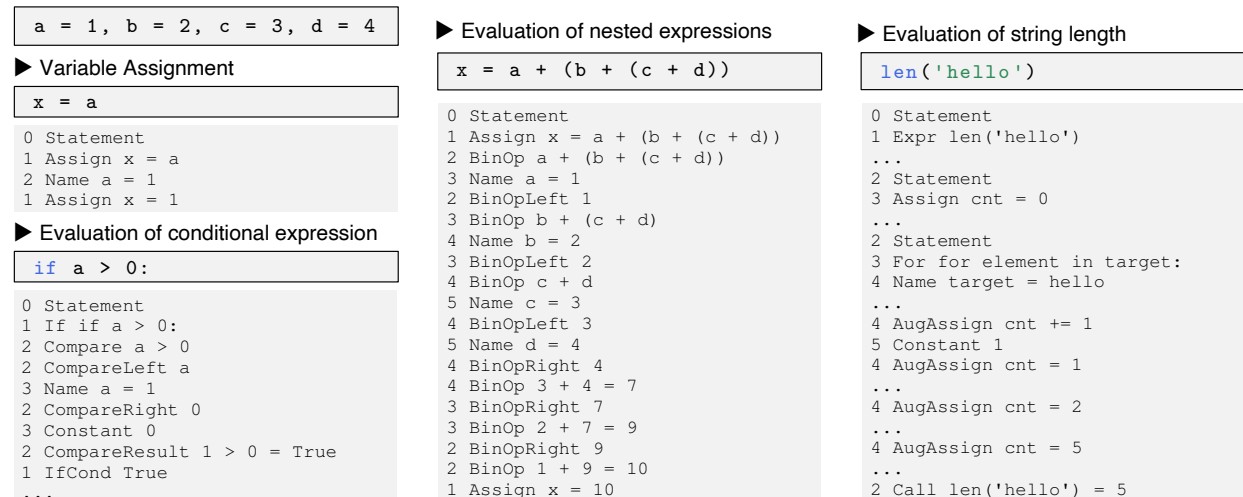

Figure 4: Examples of PyTracify's fine-grained execution traces for common Python constructs. In all examples, the variables are assumed to be initialized as `a=1, b=2, c=3, d=4`, as indicated in the top-left.

that generates execution traces which evaluate each expression in a detailed and consistent manner. We expect this enables LMs to solidify understanding of minimal execution steps and to consistently accumulate reasoning steps in a way aligned with the actual code execution steps. Below, we explain dataset construction (Section 4.1, 4.2) and design of PyTracify (Section 4.3).

## 4.1 ET-CoT dataset construction

We started with creating a high-quality dataset of code, input, and output pairs. While some prior work generated multiple input–output pairs from the same code (Li et al., 2025b), we prioritized data diversity to encourage generalization, and therefore constructed a dataset without reusing code.

Specifically, our dataset comprises five sources: **AtCoder** and **LeetCode** subsets from the **Nan-Do** dataset (Nan-Do, 2023), representing competitive programming problems; **APPS** (Hendrycks et al., 2021) and **MBPP** (Austin et al., 2021) as general code-generation tasks; **PyX**, a curated mixed dataset introduced in prior work (Ding et al., 2024a); and finally, the **Custom Dataset** described in Section 4.2.

To ensure that programs are executable, we imposed a 5-second execution limit and retained only those that completed successfully. We further removed the top 20% of samples with the longest traces, based on the trace length defined by PyTracify, to filter out excessively long cases. In addition, to ensure that our evaluation datasets were free of overlapping problems, we applied the decontamination script from Open-R1 (Face, 2025) using 8-gram matching, following Sections 3.3 and A.3 of Jain et al. (2025). After these procedures, we obtained 127,413 samples of code, input, and output. The dataset composition is summarized in Table 1.

Table 1: Composition of the dataset sources.

| Dataset Source | #Samples | Pct. (%) |
|---|---|---|
| Nan-Do | 50,426 | 39.6 |
| *AtCoder contests* | *(33,290)* | |
| *LeetCode contests* | *(17,136)* | |
| APPS | 25,908 | 20.3 |
| Custom Dataset | 38,879 | 30.5 |
| *String Functions* | *(12,000)* | |
| *Token Character-Length* | *(26,879)* | |
| PyX | 10,958 | 8.6 |
| MBPP | 1,242 | 1.0 |
| **Total** | **127,413** | **100.0** |

## 4.2 Custom dataset

As discussed in Section 3, LMs exhibited limitations even in understanding of basic operations. In particular, LMs process text token by token and tend to struggle with string operations and position identification

within strings and lists. Although CER benchmarks contain such problems, they are relatively scarce in the publicly available code datasets on which we primarily rely as data sources. To address this issue, we developed custom problem sets as follows:

**String functions dataset.** To enhance the capability for string manipulation, we created a dataset focusing on eight string functions: `slicing`, `replace`, `rpartition`, `find`, `join`, `len`, `removeprefix`, and `rstrip`. Although these functions are simple for both humans and computers, they are known to frequently mislead LMs (Gu et al., 2024b). For each function, we generated 1,500 samples by applying it to randomly generated strings of length 3–20 characters, resulting in a total of $1{,}500 \times 8 = 12{,}000$ samples. We provide an example in Appendix C.2.

**Token character-length dataset.** Another factor contributing to these limitations is that LMs do not reliably predict character length because they process text through subword tokenization rather than at the character level. To give models a correct understanding of string length, using Llama3.1-8B-Instruct (Dubey et al., 2024) as an example, we identified vocabulary items for which the model failed to predict the correct length. From this, we generated 26,879 samples of tasks to predict the output of `len`.

Section 5.4.2 reports the performance decrease of removing these custom datasets, supporting their contribution to the code execution reasoning performance.

### 4.3 Generating execution traces with PyTracify

To convert these problems into execution traces, we designed a custom Python interpreter, *PyTracify*. We configured it to decompose and record expression evaluations down to the level of atomic operations. As we show in Figure 5, each trace entry is represented as a triplet consisting of the **nest depth**, **mnemonic**, and **operation**. The **nest depth** field encodes the depth of the current call or loop nesting so the model can easily track recursion and control flow. **Mnemonic** identifies the type of operation being executed and follows the naming conventions of Python's AST node types. **Operation** records the specific code fragment being evaluated or the outcome of evaluation, such as a computed value or a state update.

Figure 4 on the previous page shows examples of Python snippets and their corresponding execution traces. The examples illustrate that a single line of code expands into multiple lines in the execution trace. Specifically,

- **Variable assignment**: Before actually assigning the new variable, we emit the expression and evaluate the right-hand side.

- **Evaluation of conditional expressions**: When evaluating a comparison, we first indicate the start of the comparison with `Compare`, then examine the left and right operands using `CompareLeft` and `CompareRight`, and finally output the result with `CompareResult`. To resolve the values of `CompareLeft` and `CompareRight`, the trace may further nest their evaluations.

- **Evaluation of nested expressions**: PyTracify thoroughly expands nested expressions. It first descends to the deepest level and then evaluates each subexpression as it unwinds. Simply logging intermediate variables once per line of code does not decompose a line even if that line is complicated. In contrast, we always decompose the computation into the level of atomic operations.

- **Evaluation of string length**: Because LMs often struggle with applying `len` to strings due to tokenization, we override `len` so that PyTracify counts characters one by one.

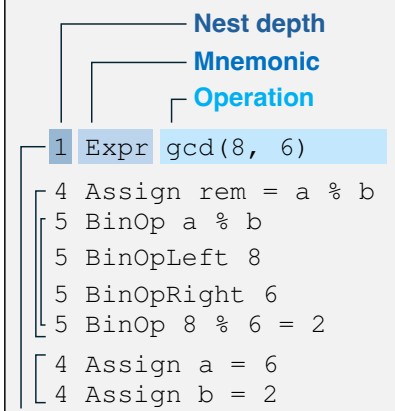

```
# Euclid's algorithm for GCD
def gcd(a, b):
    while b != 0:
        rem = a % b
        a, b = b, rem
    return a

gcd(8, 6)
```

**Nest depth**
**Mnemonic**
**Operation**

```
1 Expr   gcd(8, 6)

4 Assign rem = a % b
5 BinOp a % b
5 BinOpLeft 8
5 BinOpRight 6
5 BinOp 8 % 6 = 2

4 Assign a = 6
4 Assign b = 2
```

Figure 5: PyTracify trace format.

See Appendix A.1 for more examples. In our implementation of PyTracify, it first parses source code with the `ast` module, and then evaluates statements and expressions recursively while maintaining explicit stack frames. Its evaluation rules largely mirror those of CPython.

Because the whole pipeline executes programs without using any LLM, generation of execution traces requires minimal computational cost. In our case, by launching 100 parallel processes, we generated the entire ET-CoT dataset of 127,413 samples in just 7 minutes.

## 5    Experimental results

In this section, we present the fine-tuning results obtained with ET-CoT and compare the performance with other training methods and models. Based on the results in Section 3, we selected several representative 8B-class models from non-reasoning models for which we expect ET-CoT to improve performance: Llama3.1-8B-Instruct (Dubey et al., 2024), Qwen2.5-7B-Instruct (Qwen team, 2025), and OLMo2-7B-Instruct (OLMo team et al., 2024). To examine effectiveness across a wider range of model sizes, we additionally included Qwen2.5-0.5B-Instruct (Qwen team, 2025), Llama3.2-3B-Instruct (Dubey et al., 2024), and Qwen2.5-3B-Instruct (Qwen team, 2025). Furthermore, we added two reasoning models, Qwen3-8B (Yang et al., 2025) and OLMo3-8B-Think (Ettinger et al., 2025).[1] All models were trained on the 127k ET-CoT dataset for 4 epochs using AdamW (Loshchilov & Hutter, 2017) as an optimizer. Hyperparameters were $\beta_1$=0.9, $\beta_2$=0.95, and $\epsilon$=1e-8, and the learning rate followed a cosine decay schedule from $2 \times 10^{-5}$ to $4 \times 10^{-6}$. The batch size was 64 and context length was 8192 tokens, except for OLMo2-7B-Instruct, for which we used 4096 due to its inherent limit. The prompt format used during training can be found in Appendix C.1.

### 5.1    Performance on code execution reasoning benchmarks

We selected five datasets as benchmarks for evaluating code execution reasoning. Following SemCoder (Ding et al., 2024a), we chose CRUXEval-O (Gu et al., 2024b) and LiveCodeBench (execution, LCB-Exec) (Jain et al., 2025) from the benchmarks specifically designed for CER. In addition, we used code, input, and output pairs from HumanEval (Chen et al., 2021) as a general code benchmark and the competitive programming datasets Aizu and HackerEarth (Li et al., 2022), each filtered beforehand (see Appendix C.3).

We report the full results in Table 2 in the next page. In the upper half of the table, we report the results of the base models and ET-CoT, as well as direct-output fine-tuning (SFT; for 7–8B models) and Group Relative Policy Optimization (GRPO) (Shao et al., 2024) (for Llama3.1 and Qwen2.5). While SFT utilized exactly the same hyperparameters as those for ET-CoT, details of the GRPO training are found in Appendix C.1. In addition, for Llama3.1-8B and Qwen2.5-7B, we also report results from models obtained by fine-tuning these base models on datasets containing high-quality CoT traces, as well as from models obtained by distilling a reasoning model into these base models, for fair comparison. Specifically, as representative examples of fine-tuning on CoT data, we use BAAI/Infinity-Instruct-3M-0625-Llama3-8B, which is obtained by fine-tuning Llama3-8B with Infinity-Instruct (Li et al., 2025a), and OpenThinker3-7B, which applies reasoning traces from QwQ-32B to Qwen2.5-7B (Guha et al., 2025). As distillation baselines, we use models distilled from DeepSeek-R1 (DeepSeek-AI, 2025). The lower half of the table presents the performance of other model series, many of which were also used as baselines in SemCoder (Ding et al., 2024a).

The prompt templates used for evaluation are provided in Appendix C.4. Also, in the appendix, we provide several additional results that complement Table 2. Specifically, Appendix C.5 compares the token/character length during inference with those of a reasoning model (Qwen3-8B) and a non-reasoning model (Llama3.1-8B-Instruct). Appendix C.6 presents the results of applying ET-CoT starting from Qwen2.5-Coder-7B-Instruct, which achieve performance comparable to Qwen2.5-7B-Instruct. Furthermore, Appendix C.7 analyzes how much execution traces generated by ET-CoT models match the ground-truth traces of PyTracify.

**ET-CoT effectively converts non-reasoning models into strong code interpreters.**   From the upper part of Table 2 (ET-CoT vs. Baselines), we observe that ET-CoT substantially improves the code execution reasoning ability of language models that originally exhibited limitations even in the understanding of basic operations (Section 3). For example, ET-CoT raises Qwen2.5-7B-Instruct to 70.0% on CRUXEval-O and 88.3% on LiveCodeBench (execution), exceeding the performance of the official code-specific model Qwen2.5-Coder-7B-Instruct. This suggests that training LMs specifically for code execution reasoning with

---

[1]We note that Ettinger et al. (2025) appeared after the submission of this paper.

Table 2: Performance on five code-execution benchmarks (pass@1). ET-CoT consistently improves code execution reasoning ability across model–dataset pairs, except for reasoning models.

| Model | Size | CRUXEval-O | LCB-Exec | HackerEarth | Aizu | HumanEval | Average |
|---|---|---|---|---|---|---|---|
| **ET-CoT vs Baselines** | | | | | | | |
| Llama3.1-8B-Inst (Baseline) | 8B | 31.5 | 31.1 | 43.5 | 40.3 | 43.8 | 38.0 |
| Llama3.1-8B-Inst + SFT | 8B | 36.1 (+4.6) | 32.9 (+1.8) | 46.9 (+3.4) | 38.5 (-1.8) | 38.3 (-5.5) | 38.5 (+0.5) |
| Llama3.1-8B + Infinity-Inst [Li et al., 2025a] | 8B | 46.0 (+14.5) | 38.4 (+7.3) | 56.7 (+13.2) | 43.8 (+3.5) | 59.0 (+15.3) | 48.8 (+10.8) |
| DS-R1-Distill-Llama-8B | 8B | 66.1 (+34.6) | 82.5 (+51.4) | 72.5 (+28.9) | 67.7 (+27.4) | 77.4 (+33.6) | 73.2 (+35.2) |
| Llama3.1-8B-Inst + GRPO | 8B | 28.2 (-3.3) | 30.3 (-0.8) | 46.9 (+3.4) | 37.8 (-2.5) | 43.2 (-0.6) | 37.3 (-0.7) |
| Llama3.1-8B-Inst + ET-CoT | 8B | **67.8** (+36.3) | **88.5** (+57.4) | **75.6** (+32.1) | **71.2** (+30.9) | 71.8 (+28.0) | **75.0** (+37.0) |
| Llama3.2-3B-Inst (Baseline) | 3B | 39.9 | 54.5 | 56.7 | 50.0 | 55.1 | 51.2 |
| Llama3.2-3B-Inst + ET-CoT | 3B | **61.5** (+21.6) | **81.2** (+26.7) | **82.3** (+25.6) | **80.1** (+30.1) | **75.4** (+20.3) | **76.1** (+24.9) |
| Qwen2.5-7B-Inst (Baseline) | 7B | 42.4 | 52.8 | 64.6 | 60.2 | 66.1 | 57.2 |
| Qwen2.5-Coder-7B-Inst [Hui et al., 2024] | 7B | 65.5 (+23.1) | 64.9 (+12.1) | 75.3 (+10.7) | 69.0 (+8.8) | 77.7 (+11.6) | 70.5 (+13.3) |
| Qwen2.5-7B-Inst + SFT | 7B | 44.8 (+2.4) | 57.6 (+4.8) | 54.8 (-9.8) | 45.6 (-14.6) | 55.7 (-10.4) | 51.7 (-5.5) |
| OpenThinker3-7B [Guha et al., 2025] | 7B | 56.9 (+14.5) | 78.9 (+26.1) | 71.1 (+6.5) | 70.8 (+10.6) | 72.3 (+6.2) | 70.0 (+12.8) |
| DS-R1-Distill-Qwen-7B | 7B | 64.6 (+22.2) | 81.6 (+28.8) | 74.4 (+9.8) | 72.1 (+11.9) | **81.7** (+15.6) | 74.9 (+17.7) |
| Qwen2.5-7B-Inst + GRPO | 7B | 43.0 (+0.6) | 54.2 (+1.4) | 60.5 (-4.1) | 56.4 (-3.8) | 65.3 (-0.8) | 55.9 (-1.3) |
| Qwen2.5-7B-Inst + ET-CoT | 7B | **70.0** (+27.6) | **88.3** (+35.5) | **77.5** (+12.9) | **76.1** (+15.9) | 71.6 (+5.5) | **76.7** (+19.5) |
| Qwen2.5-3B-Inst (Baseline) | 3B | 43.3 | 38.8 | 53.1 | 48.7 | 56.0 | 48.0 |
| Qwen2.5-3B-Inst + ET-CoT | 3B | **59.4** (+16.1) | **68.5** (+29.7) | **77.3** (+24.2) | **73.5** (+24.8) | **67.1** (+11.2) | **69.1** (+21.2) |
| Qwen2.5-0.5B-Inst (Baseline) | 0.5B | 4.8 | 5.0 | 9.3 | 3.1 | 9.8 | 6.4 |
| Qwen2.5-0.5B-Inst + ET-CoT | 0.5B | **39.3** (+34.5) | **62.0** (+57.0) | **63.5** (+54.2) | **56.6** (+53.5) | **53.4** (+43.6) | **55.0** (+48.6) |
| OLMo2-7B-Inst (Baseline) | 7B | 25.0 | 24.6 | 35.4 | 35.0 | 40.0 | 32.0 |
| OLMo2-7B-Inst + SFT | 7B | 27.5 (+2.5) | 30.5 (+5.9) | 43.3 (+7.9) | 28.8 (-6.2) | 36.7 (-3.3) | 33.3 (+1.3) |
| OLMo2-7B-Inst + ET-CoT | 7B | **54.3** (+29.3) | **75.4** (+50.8) | **68.8** (+33.4) | **66.4** (+31.4) | **60.6** (+20.6) | **65.1** (+33.1) |
| Qwen3-8B (Baseline) | 8B | 65.0 | 90.8 | 77.0 | 75.2 | **77.8** | 77.2 |
| Qwen3-8B + SFT | 8B | 51.4 (-13.6) | 53.4 (-37.4) | 57.6 (-19.4) | 48.2 (-27.0) | 60.2 (-17.6) | 54.2 (-23.0) |
| Qwen3-8B + ET-CoT | 8B | **73.9** (+8.9) | **91.2** (+0.4) | **79.5** (+2.5) | **77.0** (+1.8) | 70.8 (-7.0) | **78.5** (+1.3) |
| OLMo3-8B (Baseline) | 8B | **85.5** | **94.5** | **88.5** | **89.5** | **95.7** | **90.7** |
| OLMo3-8B + ET-CoT | 8B | 68.0 (-17.5) | 87.9 (-6.6) | 88.1 (-0.4) | 84.2 (-5.3) | 84.7 (-10.9) | 82.6 (-8.2) |
| **Other Model Series** | | | | | | | |
| DeepSeek-Coder-V2-Lite [Zhu et al., 2024] | 16B | 44.6 | **68.5** | 64.9 | 54.9 | **67.9** | 60.2 |
| DeepSeek-Coder-V2-Lite-Inst [Zhu et al., 2024] | 16B | 42.1 | 46.6 | 64.9 | 56.2 | 63.0 | 54.6 |
| StarCoder2 [Lozhkov et al., 2024] | 15B | 44.4 | 39.2 | 63.8 | 55.3 | 58.9 | 52.3 |
| StarCoder2-Inst [Lozhkov et al., 2024] | 15B | 45.6 | 40.5 | 64.6 | **57.5** | 57.5 | 53.1 |
| CodeLlama-Python [Roziere et al., 2023] | 13B | 37.9 | 30.3 | 52.8 | 46.9 | 52.5 | 44.1 |
| CodeLlama-Inst [Roziere et al., 2023] | 13B | 39.5 | 30.9 | 51.1 | 43.4 | 51.8 | 43.3 |
| StarCoder2 [Lozhkov et al., 2024] | 7B | 35.0 | 32.8 | 53.4 | 45.1 | 47.7 | 42.8 |
| CodeLlama-Python [Roziere et al., 2023] | 7B | 36.8 | 32.2 | 50.8 | 45.1 | 49.0 | 42.8 |
| CodeLlama-Inst [Roziere et al., 2023] | 7B | 36.4 | 31.7 | 51.1 | 44.2 | 45.3 | 41.8 |
| DeepSeek-Coder [Guo et al., 2024] | 6.7B | 39.6 | 40.9 | 58.4 | 48.2 | 51.7 | 47.8 |
| DeepSeek-Coder-Inst [Guo et al., 2024] | 6.7B | 40.1 | 22.1 | 47.8 | 37.2 | 47.0 | 38.8 |
| Magicoder-DS [Wei et al., 2024] | 6.7B | 36.4 | 40.3 | 51.4 | 43.8 | 48.6 | 44.1 |
| Magicoder-S-DS [Wei et al., 2024] | 6.7B | 32.9 | 30.1 | 46.1 | 37.6 | 43.5 | 38.0 |
| SemCoder [Ding et al., 2024a] | 6.7B | 61.5 | 58.2 | 62.9 | 53.1 | 63.0 | 59.8 |
| SemCoder-S [Ding et al., 2024a] | 6.7B | **63.1** | 49.3 | **66.3** | 57.1 | 66.1 | **60.4** |

ET-CoT can be more effective for improving CER performance than fine-tuning them on large, general-purpose code datasets. Substantial gains are also observed for smaller models such as Qwen2.5-0.5B-Instruct and the 3B models, indicating that ET-CoT is effective across a wide range of model sizes. The exception is reasoning models, particularly OLMo3 (Ettinger et al., 2025), which, although released after the submission of this paper, saturates the benchmark. We discuss this point later.

Next, we compare these results with those in the lower half of the table. When comparing the results of the SemCoder family with the ET-CoT results for the 7–8B models, except for a few OLMo2 cases, ET-CoT outperforms the SemCoder family, which has been representative among existing fine-tuning methods using intermediate reasoning steps described in natural language. Also, the comparison with the DeepSeek-Coder-

V2-Lite series provides an example where 8B-class models with ET-CoT can be comparable to those of 16B coding models.

Overall, these results indicate the effectiveness of ET-CoT for enhancing CER performance. These gains suggest the benefit of providing highly granular and consistent reasoning steps that guide the model through the underlying computational process.

**Limitations of the training without intermediate reasoning steps.** SFT on input-output pairs and GRPO yielded only limited improvements. We think that the failure of SFT reflects a fundamental limitation of approaches without intermediate reasoning steps, as we showed in Section 3 that even a single operation requires chain-of-thought reasoning. This observation is consistent with Gu et al. (2024b), who showed that direct-output fine-tuning on CRUXEval was not effective in improving the score on CRUXEval. On the other hand, regarding GRPO, this failure cannot be attributed to the lack of intermediate steps as the generation length increased during training. A plausible explanation is that, when the underlying model is insufficiently capable for the task, output-based rewards become sparse and noisy, a well-known challenge in reinforcement learning (Lightman et al., 2023).

**Comparison with fine-tuning on high-quality CoT data, distillation, and reasoning models.** Fine-tuning open-source models on high-quality CoT traces, that are often generated from larger models, or transferring the behavior of reasoning models via distillation is becoming a common approach for maximizing the performance of small language models (Li et al., 2025a; Guha et al., 2025; DeepSeek-AI, 2025). The goal of ET-CoT is to demonstrate the effectiveness of accumulating fine-grained reasoning steps, rather than to compete directly with these approaches in terms of accuracy. Nevertheless, the fact that ET-CoT achieves larger gains than methods such as Infinity-Instruct (based on Llama3), OpenThinker-3 for Qwen2.5, and distilled models from DeepSeek-R1 illustrates the competitiveness of our simple approach.

On the other hand, reasoning models trained with RL already possess strong CER ability through natural language reasoning (Section 3). In such cases, the gains from ET-CoT may be limited, as observed with Qwen3-8B, or performance may even decrease, as seen with OLMo3-8B. We believe this occurs because the reasoning mode introduced by ET-CoT differs from natural language-based reasoning. When learning multiple abilities simultaneously, competition between them is commonly observed (Yu et al., 2020; Springer et al., 2025). This is particularly reasonable for OLMo3, which uses problems specifically designed for CER during mid-training, instilling a CER reasoning pattern different from that of ET-CoT (Ettinger et al., 2025). Nevertheless, we emphasize that ET-CoT can introduce deterministic, systematic, and fine-grained reasoning at much lower cost than RL, and that its simplicity helps highlight effective components for CER.

Taken together, these results indicate that the approach of systematically accumulating fine-grained reasoning steps is effective for code execution reasoning, and that such a reasoning mode can be easily introduced through fine-tuning even for small models whose understanding of basic operations is limited.

## 5.2 Improvement in understanding basic operations with ET-CoT

Having established that ET-CoT is effective on standard benchmarks, we next take a more fine-grained look at whether it addresses the limitations in understanding basic operations identified in Section 3. To examine this, we applied the same experimental setup as in Section 3 to the models fine-tuned with ET-CoT.

As shown in Table 3, ET-CoT yielded substantial improvements in basic operation understanding for non-reasoning models. For example, the accuracy of Llama3.1-8B-Instruct increased from 57.4% to 77.5%, and both Qwen2.5-7B-Instruct and OLMo2-7B-Instruct exhibit gains as well. In contrast, the performance of Qwen3-8B decreased. As discussed in the previous subsection, we think that Qwen3-8B's strong baseline performance and its inherently reasoning-oriented nature made it difficult for ET-CoT to outperform the original model with a new reasoning mode.

Table 3: Comparison of basic operation understanding before and after ET-CoT.

| Model | Original | ET-CoT |
|---|---|---|
| Llama3.1-8B-Inst | 57.4 | **77.5** |
| Qwen2.5-7B-Inst | 74.1 | **80.5** |
| Qwen3-8B | **93.0** | 82.6 |
| OLMo2-7B-Inst | 33.0 | **67.3** |

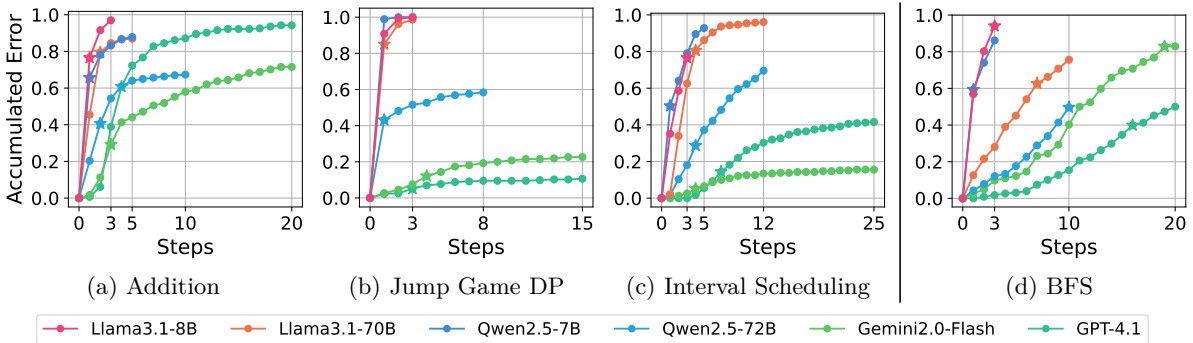

Figure 6: Step-wise cumulative error rate for iterative operations (before ET-CoT).

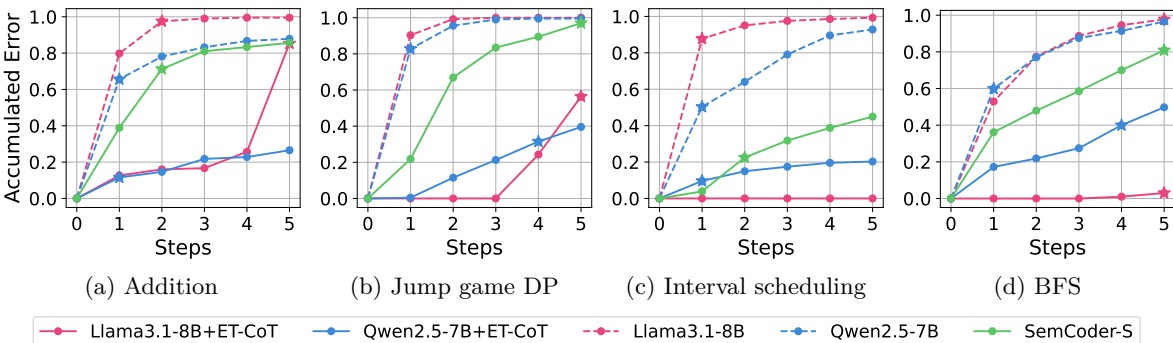

Figure 7: Step-wise cumulative error rate for iterative operations (after ET-CoT).

### 5.3 Evaluation of step-wise failure rates in iterative operations

In addition to understanding of basic operations, another fundamental component of code execution reasoning is the repeated application of a fixed operation, exemplified by the processing of a while loop. To probe this ability, prior work evaluates LMs by having them simulate algorithms with tunable step complexity (Liu et al., 2025; Chen et al., 2025; Liu & Jabbarvand, 2025), an approach also adopted in natural language reasoning (Shojaee et al., 2025). A widely observed pattern is that errors accumulate across repeated steps, eventually causing models to fail once the reasoning surpasses a certain level of complexity (La Malfa et al., 2024).

To examine how the capability to perform repeated operations changes before and after ET-CoT, we prepared four example programs specifically designed for this purpose. In selecting these algorithms, we focused on the following criteria specific to our design: (i) we designed the code such that each step consisted only of simple operations, and (ii) we inserted print statements and required the models to predict intermediate variables, thereby enforcing step-wise reasoning and identifying the incorrect steps. The four algorithms we selected are: (a) **digit-wise addition**, (b) **dynamic programming for the jump game** (where one moves from the start of the sequence to the end by steps of size one or two, minimizing the total sum of differences along the path), (c) **interval scheduling** of sorted jobs, and (d) **breadth-first search**. See Appendix D for details.

**Before ET-CoT.** The results are shown in Figure 6. We mark with a star the step at which the step-wise error rate (failure at step $t$)/(success up to step $t-1$) reaches its maximum.

Interestingly, (a), (b), and (c) exhibit a pattern in which errors were concentrated in the initial steps, followed by stable reasoning thereafter. This is in contrast to (d) BFS and to prior findings that the accumulation of repeated operations eventually leads to errors. These results indicate that, in certain cases, performance depends more on fluctuations in the model's initial state than on step-wise error accumulation. Specifically, when the model succeeds in accessing its internal knowledge of code behavior at the first step, subsequent iterations tend to be stable, whereas failure to do so can lead to an immediate breakdown. This tendency was

Table 4: Token length statistics and accuracy of different execution trace formats.

| Format | Min | Max | Mean | CRUXEval-O | LCB-Exec |
|---|---|---|---|---|---|
| **Original Trace** | 163 | 36,274 | 923.56 | 67.75 | 88.52 |
| **Removing Loop Internals** | 152 | 24,610 | 694.22 | 60.50 (-7.25) | 84.34 (-4.18) |
| **Minimal Trace** | 152 | 20,133 | 648.94 | 50.88 (-16.87) | 63.25 (-25.27) |

Table 5: Effect of dropping different subsets of the Custom Dataset (pass@1)

| Dataset | #Samples (Total) | #Samples (Custom) | CRUXEval-O | LCB-Exec |
|---|---|---|---|---|
| **Full data** | 127,413 | 38,879 | 67.75 | 88.52 |
| **No Token Character-Length** | 100,534 | 12,000 | 66.25 (-1.50) | 88.41 (-0.11) |
| **No String Functions** | 115,413 | 26,879 | 62.88 (-4.87) | 87.27 (-1.25) |
| **No Custom Dataset** | 88,534 | 0 | 57.49 (-10.26) | 85.38 (-3.14) |

observed across both open and commercial models, and Appendix D provides extensive results demonstrating the robustness of this finding.

**After ET-CoT.** The results for Llama3.1-8B and Qwen2.5-7B with $n = 5$ are presented in Figure 7. Full results are provided in Appendix D. Alongside the original and ET-CoT models, we include a comparison with SemCoder-S (Ding et al., 2024a), a representative approach based on natural-language rationales.

Here, while the original models were unable to solve any of the tasks, ET-CoT resulted in improved accuracy for both models across all tasks. These improvements are considerably larger than those obtained with SemCoder-S. Moreover, the initial instability, a failure in the early steps despite otherwise stable later repetitions, was generally reduced. Taken together, these results provide quantitative evidence that ET-CoT improves code execution reasoning ability, and in particular, mitigates the problem of initial instability in simulation of code involving repeated operations.

## 5.4 Ablation studies

To understand which components of ET-CoT contribute to its effectiveness, we conducted an ablation study. Specifically, we conducted ablations focusing on the trace format and the dataset composition. For all experiments, we used Llama3.1-8B-Instruct and the training procedure was identical to the ET-CoT setup described at the beginning of this section. Such ablation studies, particularly those that vary the reasoning granularity by changing the trace format, are a major advantage of using synthetic training data in ET-CoT.

### 5.4.1 Effect of the execution trace format

To isolate the effect of trace granularity, we trained the model with coarser trace formats and measured how the performance changed. We experimented with three trace formats. First was the **Original Trace** format used throughout our main experiments. The details are described in Section 4.3 and Appendix A.1. Second was the **Removing Loop Internals** format, which omits repeated expression-level entries inside loops. Specifically, those subordinate to `BinOpLeft`, `BinOpRight`, `CompareLeft`, and `CompareRight` are eliminated within loop breakdowns. The third was the **Minimal Trace** format, which removes the `BinOpLeft`, `BinOpRight`, `CompareLeft`, and `CompareRight` entries themselves, which leads to omitting most of the intermediate steps. The examples comparing these trace formats are found in Appendix A.2.

As shown in Table 4, the full trace yielded the highest scores on both datasets, and performance deteriorates as traces are shortened. This suggests that accurate CER benefits from fine-grained decomposition of reasoning steps, including explicitly unrolling iterations and explicitly resolving the values of relevant variables (e.g., `BinOpLeft` and `BinOpRight`) before performing binary operations or comparisons (e.g., `BinOp`). These observations are consistent with the earlier discussion in Section 3, which demonstrated that even a single operation requires explicit chain-of-thought reasoning.

### 5.4.2  Dataset ablation

In Section 4.2, we created the Custom Dataset to address typical weaknesses of LMs in handling string functions. To quantify the contribution of it, we tested four variants of the dataset: (1) **Full Dataset**, which is the complete dataset; (2) **No String Functions**, which deletes 12,000 string function examples from the custom subset; (3) **No Token Character-Length**, which omits the token character-length subset from the custom subset; (4) **No Custom Dataset**, which removes all custom subsets (string functions and token character-length).

Table 5 shows that the full dataset yields the highest accuracy, supporting the usefulness of the two custom subsets. By taking a closer look, removing string functions has a larger impact than removing the token character-length subset. However, removing both simultaneously results in more than twice the degradation caused by removing either subset alone. This indicates that the effects of the token character-length subset and string function subset are not independent but partially overlap, likely because both involve reasoning about string manipulation.

## 6  Conclusion

This work demonstrates that decomposing code execution reasoning (CER) into extremely fine-grained steps and systematically accumulating them can significantly improve the performance of small language models whose understanding of even simple operations was previously unreliable. Specifically, we proposed ET-CoT, which uses detailed execution traces generated by our custom code interpreter as chain-of-thought rationales. ET-CoT improved CER performance across different model sizes and benchmarks, and mitigated limitations in the basic components of CER.

We note several limitations of this work. First, decomposing every line into fine-grained traces may be unnecessary when only a small portion of the program is the true bottleneck. Future work may consider enabling adaptive trace granularity or adopting curriculum learning to gradually reduce trace detail. Second, in practice, CER only becomes meaningful when integrated with other coding-related capabilities such as code generation. It remains an open challenge to enable fine-grained code execution reasoning without compromising these abilities.

Despite these limitations, we believe that focusing on a single aspect of coding ability, clarifying its difficulties, and identifying key ingredients for addressing them, as demonstrated by results surpassing the official code-variant model, collectively serve as a useful stepping stone toward building strong coding models. For example, the insight that finer-grained reasoning steps improve code execution reasoning may also guide the choice of reasoning-step granularity when preparing natural-language reasoning data for fine-tuning to improve CER ability.

### Acknowledgments

Kazusato Oko and Kenshin Yamauchi contributed to this work during internships at Sakana AI. We thank Taishi Nakamura, Kou Misaki, and Llion Jones for helpful discussions.

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

# A  Details of PyTracify's trace format

## A.1  Detailed description of PyTracify's trace format

In addition to the explanations provided in the main text, this section presents more examples of PyTracify traces along with a more detailed description of the trace syntax. Figure 8 shows a sample Python program and its corresponding PyTracify trace. In all examples shown in the figure, the variables are assumed to be initialized as `a=1, b=2, c=3, d=4, y=0`, as indicated in the top-left corner.

**Variable Assignment.**  In the example of assigning the constant `1` to `x`, PyTracify first identifies the statement to be executed (`Assign x = 1`), then evaluates the constant value (`Constant 1`), and finally issues the assignment statement again (`Assign x = 1`).

In the example of assigning the variable `a = 1` to `x`, PyTracify first identifies the statement to be executed (`Assign x = a`), then evaluates the right-hand-side value (`Name a = 1`), and finally performs the concrete assignment with the resolved value (`Assign x = 1`).

**Binary Operation.**  In the example of assigning `a + 1` to `x`, the binary operation `a + 1` and the assignment to `x` are nested. Within the binary operation, PyTracify first identifies the expression to be evaluated (`BinOp a + 1`), then resolves the left-hand side through `Name a = 1` and `BinOpLeft 1`. The right-hand side is resolved via `Constant 1` and `BinOpRight 1`. Once both `BinOpLeft` and `BinOpRight` are obtained as concrete values, it computes the result `BinOp 1 + 1 = 2`. In this way, a binary operation evaluates its left and right operands separately until each reduces to a constant before performing the operation itself. This design allows the core computation step (e.g., `1 + 1 = 2`) to be singled out in an extremely simple form.

**Comparison.**  Comparisons are handled analogously. Both sides are evaluated to constants prior to the comparison. In the shown example, after declaring `Compare a > b`, PyTracify evaluates `CompareLeft` and `CompareRight`, and finally obtains the result through `CompareResult 1 > 2 = False`. As with binary operations, this approach isolates the essential comparison step in a simple and explicit form.

We remark that bare variables are evaluated through the `Expr` construct.

**Evaluation of conditional expression.**  A conditional expression internally invokes a comparison (or another form of condition). In this example, after identifying the structure of the if-statement (`If if a > 0`), PyTracify evaluates the condition `a > 0`. Once the comparison is resolved as `True` through `CompareResult 1 > 0 = True`, it returns `IfCond True`.

**Evaluation of nested expressions.**  As noted in the main text, nested expressions are expanded sequentially. This enables a much finer-grained evaluation of nested expressions than approaches that simply log intermediate variables on a per-line basis. In this example, PyTracify begins with `2 BinOp a + (b + (c + d))`, then expands the inner expressions through `3 BinOp b + (c + d)` and `4 BinOp c + d`. After obtaining the intermediate results, it unwinds the nesting through `4 BinOp 3 + 4 = 7`, `3 BinOp 7 + 2 = 9`, and `2 BinOp 9 + 1 = 10`, and finally assigns the resolved value with `Assign x = 10`.

**While loop.**  A while loop is executed by repeatedly evaluating the condition within `WhileCond`...`WhileCondResult`, and performing the computations under `WhileBody`. Note that `AugAssign` in this example represents an augmented assignment, and functions analogously to the variable assignment operations discussed earlier.

**For loop.**  A for-loop, as in this example, repeats the definition of the loop variable (e.g., `Name i = 1`) and the computations within the loop body. Before entering the loop, all elements of the list are individually enumerated as `List0 1`, `List1 2`, and so on.

**Evaluation of string length.**  PyTracify overrides Python's built-in function `len` as shown in Listing 6. This is because LMs typically process strings as sequences of subword tokens, making them especially weak

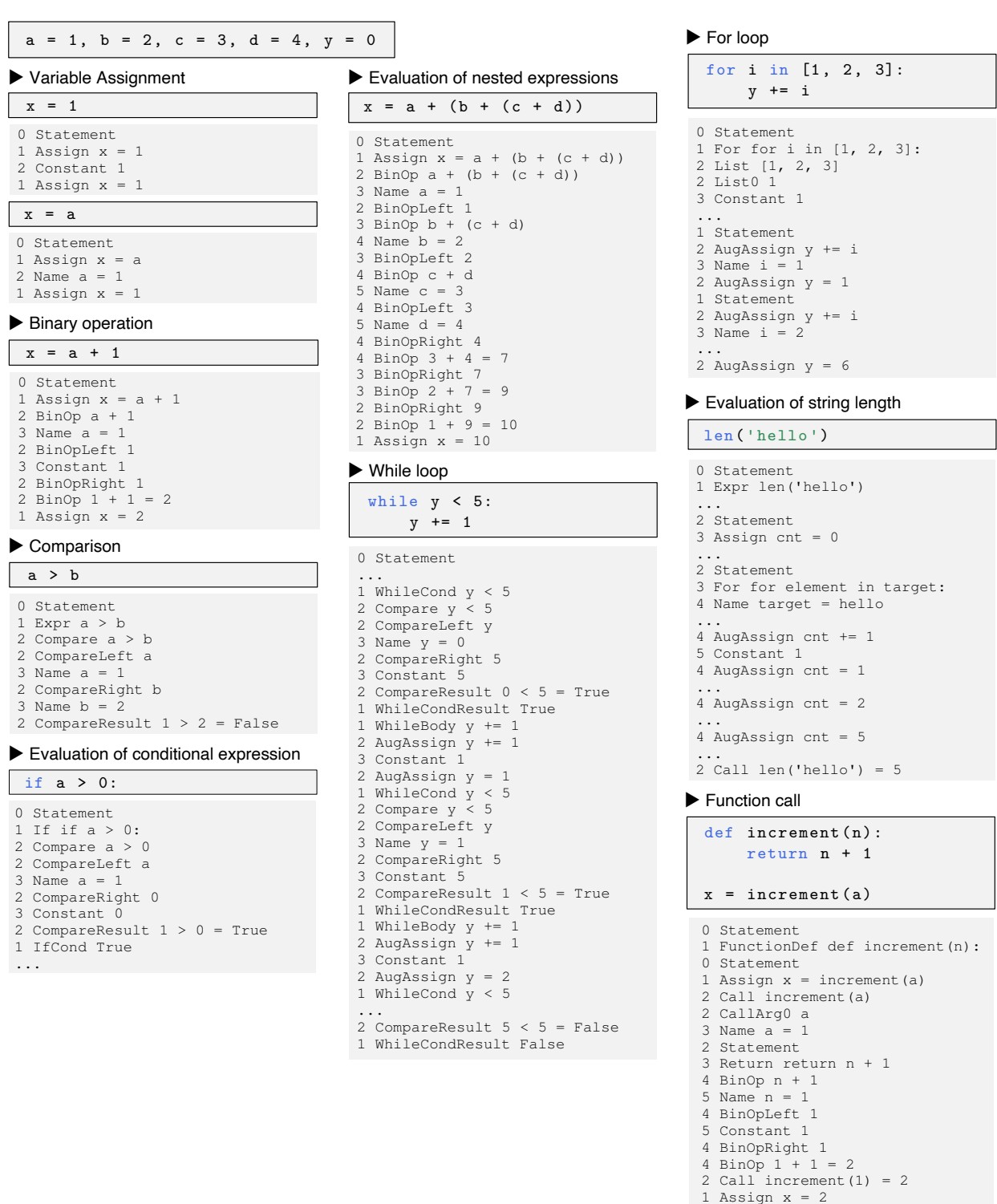

Figure 8: More Examples of PyTracify's fine-grained execution traces for common Python constructs.

at applying `len` to strings. By overriding `len` to expose the per-character iteration in the execution trace, PyTracify makes character-level counting more accessible for the model to learn. In the example in Figure 8, the string 'hello' is decomposed into individual characters, and `cnt` is updated one character at a time.

**Function call.** Function calls are handled as follows. When a function definition is encountered, its name is recorded using `FunctionDef`, while the actual execution is performed only when the function is invoked. In this example, the call `Call increment(a)` triggers the evaluation of its argument through `CallArg0 a` and `Name a = 1`. The function body is then executed, and finally `Call increment(1) = 2` is returned and assigned to `x`.

**Listing 6    Override of `len` to expose character-level iteration**

```python
def len(target: object) -> int:
    cnt = 0
    for element in target:
        cnt += 1
    return cnt
```

## A.2    Different trace formats used for the ablation study

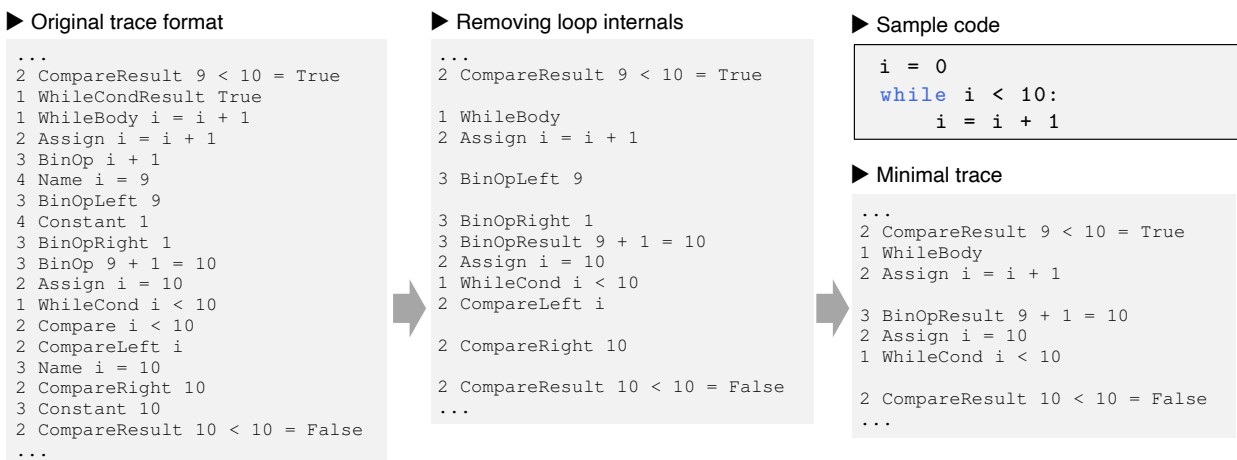

Figure 9: Examples of different styles of execution traces.

In Figure 9, we show an example of the three trace formats. The Original Trace format is most detailed, and was used as ET-CoT. The Removing Loop Internals format intends to remove redundancy in the for and while loops. This omits repeated expressions which are subordinate to `BinOpLeft`, `BinOpRight`, `CompareLeft`, and `CompareRight` within loop breakdowns. The Minimal Trace format, which intends to omit most of the intermediate reasoning steps, removes the `BinOpLeft`, `BinOpRight`, `CompareLeft`, and `CompareRight` entries themselves.

Figure 9 shows that the three trace formats decrease in length in order. The blank lines indicate the parts that were removed compared to the trace at the previous level.

## A.3    Supplementary discussion on implementation differences between PyTracify and CPython

PyTracify faithfully reproduces CPython's surface syntax in almost all respects. The only notable difference lies in how assignments inside nested functions determine the scope of the target variable.

In assignment binding, CPython treats an assignment inside an inner function as creating a new local variable unless the name is explicitly declared with nonlocal or global. In contrast, PyTracify adopts a simpler rule. Specifically, if the assigned name already exists in the nearest enclosing function scope, the assignment updates that enclosing binding. Otherwise, the assignment creates a local variable, mirroring CPython. In effect, PyTracify behaves as though the variable had been explicitly marked as belonging to the enclosing function scope, or as though nonlocal were implicitly applied. This design choice simplifies trace generation because PyTracify does not need to reproduce Python's compile-time scope analysis, but the simple programs used in this work are not unaffected by this.

Listing 7 shows a (rare) example where the behavior differs. Under CPython, the variable `x` assigned inside `inner()` is treated as a new local binding unless `nonlocal x` is declared, so `outer()` prints 'enclosing'. In contrast, PyTracify updates the enclosing `x` by default, so `outer()` prints 'local'.

**Listing 7    Example of Differences in Assignment Binding Within Nested Functions**

```python
x = 'global'

def outer():
    x = 'enclosing'
    def inner():
        x = 'local'
        print(x)   # CPython: 'local'  | PyTracify: 'local'
    inner()
    print(x)       # CPython: 'enclosing'
                   # PyTracify: 'local'  (inner updated the enclosing x)

outer()
print(x)           # CPython: 'global'  | PyTracify: 'global'
```

## B    Details of the experiments in Sections 3 and 5.2

In Section 3 and Section 5.3, we introduced a custom test designed to evaluate the ability of LMs to perform basic operations. This appendix section describes the data generation and evaluation pipeline in detail.

### B.1    Data generation pipeline

The problems were constructed based on the CRUXEval (Gu et al., 2024b). CRUXEval consists of 800 pairs of 3–13 line functions, inputs, and corresponding outputs. A sample problem is shown in Listing 8.

**Listing 8    Sample problem from CRUXEval**

Function:
```python
def f(s, o):
    if s.startswith(o):
        return s
    return o + f(s, o[-2::-1])
```
Input:
```python
s = 'abba'
o = 'bab'
```
Output:
```python
'bababba'
```

For each original problem, we constructed a simplified problem aimed at assessing the model's ability to execute a single basic operation. We asked Gemini 2.5 Pro (Comanici et al., 2025) to conduct the following process. First, extract the line deemed most essential from each original function. Next, create a minimal function containing only that line (with minimal scaffolding if strictly necessary). For each such function, generate input–output pairs, ensuring that the new function's behavior matched the execution of the extracted line in the original function. The system and user prompts for this are shown in Listing 9.

**Listing 9    Sample prompt for Gemini 2.5 Pro to transform the problem**

System prompt:
```
You are a dataset transformation assistant.

You will work with problems from Cruxeval, a code execution dataset:
```

```
- Each problem is (code, input, output).
- Given code and input, the original task is to predict the execution output.

Before transforming, simulate the original code step by step to locate the first
  ↪ moment when the single most essential line is executed and to capture the
  ↪ exact variable state at that moment.

Transformation rules:
1) Choose exactly one "most essential" line.
2) Write a minimal function that contains only that line (plus minimal
  ↪ scaffolding if strictly necessary).
3) Creating the Input: use the exact variable state when that line first executes
  ↪  as the new inputs.
   Write them in the same comma-separated style shown in the examples.
   Keep it minimal but sufficient to execute the line once.
4) Creating the Output: the return value produced by the new simplified function
  ↪ after executing the essential line exactly once.

You may output reasoning before the transformed data, but always end with the
  ↪ following format:

code:
<transformed code>

input:
<transformed input>

output:
<transformed output>

---

### Example 1

Original:
code:
def f(text, value):
    text_list = list(text)
    text_list.append(value)
    return ''.join(text_list)

input:
'bcksrut', 'q'

output:
'bcksrutq'

Transformed:
code:
def f(text_list, value):
    text_list.append(value)
    return text_list

input:
text_list = ['b', 'c', 'k', 's', 'r', 'u', 't']
value = 'q'

output:
['b', 'c', 'k', 's', 'r', 'u', 't', 'q']

---

### Example 2

Original:
code:
def f(names):
    numberOfNames = 0
```

```
    for i in names:
        if i.isalpha():
            print(i, numberOfNames)
            numberOfNames += 1
    return numberOfNames

input:
['sharron', 'Savannah', 'Mike Cherokee']

output:
2

Transformed:
code:
def f(name):
    if name.isalpha():
        return 1
    else:
        return 0

input:
name = 'sharron'

output:
1
```

User prompt:
```
Transform the following Cruxeval problem according to the System instructions.

code:
def f(s, o):
    if s.startswith(o):
        return s
    return o + f(s, o[-2::-1])

input:
'abba', 'bab'

output:
'bababba'
```

Among the generated samples, about ten were either non-executable or had mismatched input–output pairs. For these cases, we applied minimal manual fixes to these cases. As a result, we obtained 800 simplified functions together with correct input–output pairs. Listing 10 shows how the example problem in Listing 8 was transformed, along with the complete output generated from Gemini 2.5 Pro.

**Listing 10   Example of a transformed problem and the full model output**

Function:
```
def f(o):
    return o[-2::-1]
```
Input:
```
o = 'bab'
```
Output:
```
'ab'
```
Complete output:
```
`f` is a recursive function. Let's trace its execution with the given inputs.

1.  **`f(s='abba', o='bab')`**:
    *   The condition `s.startswith(o)` (`'abba'.startswith('bab')`) is `False`.
    *   The `else` block is executed. It must compute `o + f(s, o[-2::-1])`.
    *   This is the first time the most essential part of the logic is reached.
      ↪ The crucial operation here is the calculation of the new value for `o`
      ↪ for the recursive call.
```

```
    *    The expression `o[-2::-1]` is evaluated. `o` is `'bab'`. The slice starts
    ↪    at the second-to-last character ('a') and goes backward to the
    ↪    beginning. The result is `'ab'`.
    *    The execution continues with a recursive call: `f('abba', 'ab')`.

2.  **`f(s='abba', o='ab')`**:
    *    The condition `s.startswith(o)` (`'abba'.startswith('ab')`) is `True`.
    *    The function returns `s`, which is `'abba'`.

3.  **Returning to the first call**:
    *    The expression from step 1 is now resolved: `'bab' + 'abba'`.
    *    The final return value is `'bababba'`.

The most essential line is `return o + f(s, o[-2::-1])`. The single most critical
  ↪   operation within that line, executed at the first opportunity, is the slice
  ↪   `o[-2::-1]`, which generates the argument for the subsequent recursive call
  ↪   and drives the algorithm forward. We will isolate this operation.

-    **Essential line chosen:** The slice operation `o[-2::-1]` from the line `
  ↪ return o + f(s, o[-2::-1])`.
-    **State at first execution:** The variables available when this slice is
  ↪ first computed are `s = 'abba'` and `o = 'bab'`. Only `o` is needed for the
  ↪ slice.
-    **Minimal function:** A function that takes `o` and returns the result of the
  ↪  slice.
-    **Input:** The value of `o` at that moment: `'bab'`.
-    **Output:** The result of `'bab'[-2::-1]`, which is `'ab'`.

code:
def f(o):
    return o[-2::-1]

input:
o = 'bab'

output:
'ab'
```

## B.2 Evaluation pipeline

On the custom dataset, we evaluated the following nine models: Llama-3.1-8B-Instruct (Dubey et al., 2024), Llama-3.1-70B-Instruct (Dubey et al., 2024), Qwen2.5-7B-Instruct (Qwen team, 2025), Qwen2.5-72B-Instruct (Qwen team, 2025), Qwen3-8B (Yang et al. (2025), think mode for CoT and explanation, non-think mode for direct prediction), Ministral-8B-Instruct-2410, OLMo-2-1124-7B-Instruct (OLMo team et al., 2024), Gemini-2.0-Flash, and GPT-4.1. When selecting the models, we categorized them into three groups. The first group consists of recent 7–8B models, including Llama-3.1-8B-Instruct, Qwen2.5-7B-Instruct, Qwen3-8B, Ministral-8B-Instruct-2410, and OLMo-2-1124-7B-Instruct. The second group contains larger models from the same families as those in the first group, namely Llama-3.1-70B-Instruct and Qwen2.5-72B-Instruct. The third group comprises widely used commercial models, for which we selected Gemini-2.0-Flash and GPT-4.1. In this group, we focused on non-thinking variants because closed-source thinking models cannot be forced to provide direct, reasoning-free answers, which would otherwise result in missing data.

For each model, we performed three types of evaluation: output prediction with and without CoT, and verification of whether the model was able to produce an accurate natural-language description of the function's behavior. For output prediction, the prompt used in the CoT setting is shown in Listing 11, while the prompt for direct output prediction is shown in Listing 12. The user prompt for direct output prediction is identical to that used in the CoT setting. Additionally, for Qwen3-8B in the direct-output setting, we prefixed the system prompt with "/nothink".

**Listing 11   Sample prompts for output prediction (with CoT)**

System prompt:

```
You are a highly capable assistant. Your task is to predict the exact value that
  ↪ the final function call evaluates to in the given Python code. Please
  ↪ describe your thought process briefly but clearly enough so that someone can
  ↪  verify your thought process, enclosed in <reason>...</reason>, and put the
  ↪ prediction of the code output in <answer>...</answer>.

Strictly follow the format:
<reason>
(thought process here)
</reason>
<answer>
(only the final output here)
</answer>
```

User prompt:
```
Predict the return value of the final function call of the code, according to the
  ↪  system instructions.

code:
def f(o):
    return o[-2::-1]

o = 'bab'

f(o)
```

---

**Listing 12    Sample prompt for output prediction (without CoT)**

---

System prompt:
```
You are a highly capable assistant. Your task is to predict the exact value that
  ↪ the final function call evaluates to in the given Python code. Return only
  ↪ the final output without any explanations. Note: strings (str) must be
  ↪ enclosed in single quotes ('') in your answer.
```

Listing 13 shows example responses from each model for the output-prediction task (with CoT), based on the problem in Listing 11. Although the task has been simplified to a single, straightforward list operation, multiple models still mispredict the output.

---

**Listing 13    Example of generated answers for output prediction (with CoT)**

---

**[GPT-4.1] (incorrect)**
Answer:
```
'ba'
```

Reasoning:
```
The function f(o) takes a sequence o and returns o[-2::-1]. In Python slice
  ↪ notation, [start:stop:step], so o[-2::-1] starts at the second-to-last
  ↪ element and goes backwards to the beginning. For o = 'bab', o[-2] is 'a', so
  ↪  the slice is from index 1 to before the beginning, stepping by -1: o
  ↪ [1::-1], which is 'ab' reversed, i.e., 'ba'.
```

**[Gemini 2.0 Flash] (correct)**
Answer:
```
'ab'
```

Reasoning:
```
The function `f` takes a string `o` as input.
It returns a slice of the string `o`. The slice `o[-2::-1]` starts from the
  ↪ second to last element (index -2), goes to the beginning of the string, and
  ↪ reverses the string.
In this case, o = 'bab'. So the slice starts from index -2 which is 'a', and goes
  ↪  to the beginning with a step of -1. So the reversed slice is 'ab'.
```

**[Llama3.1-8B] (incorrect)**
Answer:
```
'abab'
```

Reasoning:

```
Upon executing the function call f(o) with the input string 'bab', we first need
  ↪ to reverse the entire string except for the last two characters. Since the
  ↪ indexing in Python is 0-based, '-2::-1' means start from the second last
  ↪ character ('b') and move backwards, stepping by -1 each time. However, since
  ↪  we're looking to exclude the last character and only remove the second last
  ↪  digit then overwrite the first 3rd digit of the 'o' list with the rest, a
  ↪ simpler method would be to get the first 1 character and put it last here
  ↪ while the first 2 characters then get reversed leading to  reverse offirst
  ↪ three digits of string except last two, return the rest: ('a','b',''), and
  ↪ appending them at the start afteroverwriting '' leaving ('a','b','b')
  ↪ excluding the first & second part of the repeating ('b') block that were
  ↪ moved away substituting for their normal counterparts of index location {0,
  ↪ }, return the reversed list resulting sting: 'abab'.
```

**[Llama3.1-70B] (incorrect)**

Answer:

```
'aba'
```

Reasoning:

```
The function f(o) takes a string o as input and returns the substring from the
  ↪ second last character to the beginning of the string in reverse order. The
  ↪ slicing operation o[-2::-1] starts from the second last character (index -2)
  ↪ , moves backwards to the beginning of the string (step -1), and includes all
  ↪  characters in between.

In this case, the input string o is 'bab'. Applying the slicing operation, we
  ↪ start from the second last character 'a', move backwards to the beginning of
  ↪  the string, and include all characters in between. This results in the
  ↪ substring 'aba'.

However, since the slicing operation moves backwards, the resulting substring is
  ↪ in reverse order, so the actual return value of the function is 'aba'
  ↪ spelled backwards, which is 'aba' itself since it is a palindrome.
```

**[Qwen2.5-7B] (incorrect)**

Answer:

```
'b'
```

Reasoning:

```
The function `f(o)` operates on the string `o` which is 'bab'.
1. In the function, the slice `o[-2::-1]` is used. This starts from the second
  ↪ last character (i.e., the index -2, which is 'b') and goes backwards to the
  ↪ start of the string (index -3, which would be out of range, so it starts
  ↪ from 'b') step by -1.
2. So, it reverses the string from the second last character to the start, which
  ↪ results in 'b'.
```

**[Qwen2.5-72B] (correct)**

Answer:

```
'ab'
```

Reasoning:

```
The function `f` takes a single argument `o` which is expected to be a string. It
  ↪  returns a slice of the string starting from the second last character to
  ↪ the beginning of the string, in reverse order.

Given the input `o = 'bab'`:
- `o[-2]` is 'a' (the second last character).
- `o[-3]` is 'b' (the first character, accessed in reverse order).

So, the slice `o[-2::-1]` will return the substring 'ab'.
```

On the other hand, to evaluate whether the models could correctly describe the code in natural language, we employed a two-stage pipeline. First, each model was given the code and asked to provide a natural-language

explanation of its behavior. An example prompt is shown in Listing 14. The demonstration examples were taken from other items in the dataset.

Once we obtained the explanations from each model, we evaluated whether the original output could be inferred solely from the explanation and the given input. For this purpose, we used GPT-4.1, which achieved the highest accuracy in output prediction with CoT. If GPT-4.1's inferred output matched the ground-truth output, we judged the model's natural-language explanation to be correct; otherwise, it was considered incorrect. Listing 15 shows an example of the prompt used for GPT-4.1, which served as the judge.

Listing 16 presents examples of the natural-language explanations generated by each model. In these examples, GPT-4.1 was able to infer the ground-truth output in all cases. This demonstrates that, even if a language model can articulate a correct natural-language explanation of the code, it does not necessarily follow that it understands how the code actually executes on a specific input instance. The phenomenon of being able to explain a concept but failing to apply it to concrete instances has been documented as a failure mode of LMs in other domains (Mancoridis et al., 2025). Our findings show that this failure mode also manifests in the context of code understanding.

---

**Listing 14   Sample prompts used for generating natural-language explanations**

System prompt:
```
You will be given a Python function along with input and output examples. Explain
  ↪  what the function does, briefly but clearly enough so that someone could
  ↪ write the same code after reading it. Your response should consist only of
  ↪ the explanation.

---

### Example 1

function:
def f(text_list, value):
    text_list.append(value)
    return text_list

example input:
text_list = ['b', 'c', 'k', 's', 'r', 'u', 't']
value = 'q'

example output:
['b', 'c', 'k', 's', 'r', 'u', 't', 'q']

explanation:
This function takes as input a list of single-character strings called text_list
  ↪ and a single-character string called value, and it outputs a list of single-
  ↪ character strings. The output is text_list with value appended to the end.

### Example 2

function:
def f(name):
    if name.isalpha():
        return 1
    else:
        return 0

example input:
name = 'sharron'

example output:
1

explanation:
This function takes as input a string called name and outputs an integer. The
  ↪ output is 1 if the string name consists only of alphabetic characters (a-z,
  ↪ A-Z) and is non-empty; otherwise, the output is 0.
```

User prompt:

```
Generate an explanation from the following, according to the system instructions.

function:
def f(o):
    return o[-2::-1]

example input:
o = 'bab'

example output:
(omitted)
```

**Listing 15   Sample prompts for verifying natural-language explanations**

System prompt:

```
You are part of a pipeline that checks whether explanations of Python functions
  ↪ are truly correct. You are given an explanation and an input. The
  ↪ explanation is written based on a Python function. Using the explanation,
  ↪ you must predict the output of applying the function to the given input. You
  ↪  may output reasoning before the transformed data, but you must always end
  ↪ with the format shown below. Note: strings (str) must be enclosed in single
  ↪ quotes ('').

output:
<expected output>

---

### Example 1

function (you cannot see):
def f(text_list, value):
    text_list.append(value)
    return text_list

explanation:
This function takes as input a list of single-character strings called text_list
  ↪ and a single-character string called value, and it outputs a list of single-
  ↪ character strings. The output is text_list with value appended to the end.

input:
text_list = ['b', 'c', 'k', 's', 'r', 'u', 't']
value = 'q'

output:
['b', 'c', 'k', 's', 'r', 'u', 't', 'q']

### Example 2

function (you cannot see):
def f(name):
    if name.isalpha():
        return 1
    else:
        return 0

explanation:
This function takes as input a string called name and outputs an integer. The
  ↪ output is 1 if the string name consists only of alphabetic characters (a-z,
  ↪ A-Z) and is non-empty; otherwise, the output is 0.

input:
name = 'sharron'

output:
1
```

User prompt:

```
Given the explanation of a Python function and an input, generate the expected
 ↪ output of applying the function to the input, according to the system
 ↪ instructions.

explanation:
This function takes as input a string called o and outputs a string. The output
 ↪ is the substring of o starting from the second-to-last character and moving
 ↪ backward to the beginning of the string, effectively reversing all
 ↪ characters of o except for the last one.

input:
o = 'bab'
```

**Listing 16   Examples of generated explanations**

**[GPT-4.1]** (correct)

```
This function takes as input a string called o and outputs a string. The output
 ↪ is the substring of o starting from the second-to-last character and moving
 ↪ backward to the beginning of the string, effectively reversing all
 ↪ characters of o except for the last one.
```

**[Gemini 2.0 Flash]** (correct)

```
This function takes as input a string called o and returns a new string
 ↪ consisting of all characters in o except for the last one, in reverse order.
```

**[Llama3.1-8B]** (correct)

```
This function takes as input a string o and returns a substring consisting of all
 ↪  characters of o except the last one, in reverse order.
```

**[Llama3.1-70B]** (correct)

```
This function takes as input a string o and outputs a string. The output is the
 ↪ substring of o starting from the second-to-last character and moving
 ↪ backwards to the beginning of the string.
```

**[Qwen2.5-7B]** (correct)

```
This function takes as input a string o and outputs a string. The output is the
 ↪ substring of o containing all characters except the last one, taken in
 ↪ reverse order.
```

**[Qwen2.5-72B]** (correct)

```
This function takes as input a string o and outputs a string. The output is the
 ↪ substring of o starting from the second-to-last character (inclusive) to the
 ↪  beginning of the string, in reverse order.
```

# C   Details of the training and evaluation on code execution reasoning benchmarks

## C.1   Training setup details

**Prompt format for training.**   Listing 17 shows the prompt format used for ET-CoT training. In this setup, the user provides a Python code to be executed along with any required standard input values. The assistant then returns the execution trace, generated by PyTracify, enclosed within `<think>` tags, followed by the final output enclosed within `<answer>` tags.

For supervised fine-tuning (SFT), we remove the `<think>...</think>` section and train the model to output only the final answer. For GRPO, we use only the system prompt and the user prompt, without supplying execution traces.

**Listing 17   Sample prompt for ET-CoT training**

System prompt:

```
You are a highly capable assistant. Your task is to estimate the output of the
 ↪ given Python code. The reasoning process and output are enclosed within <
 ↪ think> </think> and <answer> </answer> tags, respectively, i.e., <think>
 ↪ reasoning process here </think><answer> output here </answer>
```

User prompt:

```

def fib(n):
    if n <= 1:
        return n
    return fib(n-1) + fib(n-2)

<input>
1
</input>
```

Assistant output:

```
<think>
0 Statement
1 FunctionDef def fib(n):
0 Statement
1 Expr fib(1)
2 Call fib(1)
2 CallArg0 1
3 Constant 1
2 Statement
3 If if n <= 1:
4 Compare n <= 1
4 CompareLeft n
5 Name n = 1
4 CompareRight 1
5 Constant 1
4 CompareResult 1 <= 1 = True
3 IfCond True
3 Statement
4 Return return n
5 Name n = 1
2 Call fib(1) = 1
</think>
<answer>
1
</answer>
```

**Hyperparameters for GRPO** The hyperparameters for GRPO are summarized in Table 18. We trained the model for 2,000 steps using ModelScope Swift (Zhao et al., 2024) on eight NVIDIA H200 GPUs (six for gradient computation and two as vLLM servers). Due to the computational cost, we were unable to use all 127k training examples. However, we note that whereas ET-CoT and SFT complete four epochs within a single day, the 2,000-step GRPO run required approximately three days, so we spent substantially higher computational cost on GRPO compared to ET-CoT and SFT.

## C.2 Example of the custom dataset

**Listing 19  Example of String Functions Dataset**

```

# removeprefix: Remove the specified prefix from the start of the string if
 ↪ present.
# Example: 'unhappy'.r emoveprefix('un') -> 'happy'
print('koalawatermelonslow'.removeprefix('koa'))

```

Listing 19 is an example from the **String Functions Dataset** described in Section 4.2. It shows the case of the removeprefix function, where the behavior of removeprefix is explained in comments inside the

Table 18: Core hyperparameters used for GRPO training.

| Hyperparameter | Value |
|---|---|
| Number of steps | 2000 |
| Learning rate | $1 \times 10^{-6}$ |
| Global batch size | 48 |
| Per-device batch size | 4 |
| Gradient accumulation steps | 2 |
| Number of GPUs | 6 |
| Number of generations | 8 |
| KL coefficient ($\beta$) | 0.001 |
| Max generation length | 8192 tokens |
| Temperature | 1.0 |
| Top-$p$ | 0.9 |
| Top-$k$ | 50 |

`` block, along with an example of applying the function to a random string. Similar training data is made for other functions such as `len`, `slice`, `replace`, `rpartition`, `find`, `join`, and `rstrip`.

### C.3 Preprocess of the evaluation datasets

Here, we describe how each of the HumanEval, Aizu, and HackerEarth datasets was preprocessed.

**HumanEval (708 examples).** HumanEval (Chen et al. (2021), `https://huggingface.co/datasets/openai/openai_humaneval`) is a dataset consisting of 164 hand-written functions, each accompanied by several test cases. Test cases provide input examples, from which we obtained function–input pairs.

We applied several exception-handling steps where necessary. First, some problems provide an excessively large number of inputs; for such cases, we limited the number of inputs per function to at most five. Second, for cases where a single example defined two or more functions and one function invoked another, we manually rewrote the code to consolidate it into a single function. Third, we excluded any example that failed to execute, required imports, did not finish within 0.3 seconds, or whose function definition exceeded 20 lines. After applying these steps, we obtained 708 examples, with each function appearing at most five times.

**Aizu (226 examples) and HackerEarth (356 examples).** Aizu and HackerEarth are subsets of the competitive programming datasets used for training in Li et al. (2022) (`https://huggingface.co/datasets/deepmind/code_contests`), selected because they were the only splits that did not overlap with the training data. The contest platform for Aizu is `https://judge.u-aizu.ac.jp`, and the original source of the dataset is `https://github.com/IBM/Project_CodeNet`. The contest platform for HackerEarth is `https://www.hackerearth.com`, and the original source of the dataset is `https://github.com/ethancaballero/description2code`.

In the dataset `https://huggingface.co/datasets/deepmind/code_contests`, each problem consists of a problem statement, a set of test cases, and multiple submitted solutions that pass all tests. We first filtered the dataset to retain only those problems that contained at least one solution defining a single function exactly once. For each such problem, we then used its first test case as the input and generated a single function–input pair. Then, as before, we excluded any example that failed to execute, required imports, did not finish within 0.3 seconds, or whose function definition exceeded 20 lines. As a result, we obtained 226 examples for Aizu and 356 examples for HackerEarth.

### C.4 Sample prompts for inference

We now describe the prompts used to obtain the results in Table 2. For ET-CoT, SFT, and GRPO, we used only the system and user prompts from the training template shown in Section C.1. However, for the

ET-CoT models based on Qwen2.5, OLMo2, and Llama3.2-1B, we occasionally observed that they defaulted to their standard inference mode. To ensure that the ET-CoT procedure was triggered, we therefore prefixed the prompt with the token sequence `0\n Statement`. Because there was no need to introduce diversity in the outputs, we used a temperature of 0 and top-$k$ sampling with $k = 1$. The prompts for the base models, shown in Listing 20, were based on the CoT template used in CRUXEval (Listing 21 of (Gu et al., 2024b)) and included two-shot examples.

For the remaining models, we adjusted the base-model prompt template to each model. This is because many code models are trained with specific prompt formats, and using a template aligned with a model's training setup is often necessary to obtain its best performance.

For Qwen2.5-Coder-7B-Instruct, we used the prompt shown in Listing 21, obtained from the official Qwen2.5-Coder evaluation scripts (`https://github.com/QwenLM/Qwen3-Coder/blob/main/qwencoder-eval/base/benchmarks/cruxeval/inference/prompts.py` ). For SemCoder and SemCoder-S (more precisely, SemCoder_1030 and SemCoder-S_1030), we adopted the templates from their project repository (`https://github.com/ARiSE-Lab/SemCoder/` ; see Listing 22). The prompt for Magicoder (Listing 23) was similarly obtained from the official GitHub repository (`https://github.com/ise-uiuc/magicoder` ). DeepSeek-Coder-V2-Lite was evaluated using the same prompt as the base models (Listing 20). The other models (the CodeLlama series, the StarCoder2 series, DeepSeek-Coder and DeepSeek-Coder Inst) are primarily designed for code generation rather than reasoning, so we prompted them to directly output only the final answer. For these models, we used the direct-output template taken from Listing 16 of CRUXEval (Gu et al., 2024b), see Listing 24.

With these model-specific prompt selections, we observed that among the 11 models overlapping with Table 1 of SemCoder (Ding et al., 2024a), all models except Magicoder-DS and Magicoder-S-DS achieved higher average scores on CRUXEval and LiveCodeBench than previously reported.

---

**Listing 20    Prompt templates for evaluating base models and DeepSeek-Coder-V2-Lite**

---

System prompt:

```
You are given a function and an input. Provide the output of executing the
  ↪ function on the input. Reason step by step before arriving at an answer.
  ↪ Surround the reasoning process and output with <think> </think> and <answer>
  ↪  </answer> tags, respectively.

Example 1:

def f(x):
return x

<input>
17
</input>

<think>
(reasoning process here)</think>
<answer>
17
</answer>

Example 2:

def f(x):
return x

<input>
'a'
</input>

<think>
(reasoning process here)
</think>
<answer>
```

```
'a'
</answer>
```

User prompt:

```

def f(nums):
    for i in range(len(nums)):
        if not i % 2:
            nums.append(nums[i] * nums[i + 1])
    return nums

<input>
[]
</input>
```

---

**Listing 21  Prompt templates for evaluating Qwen2.5-Coder-7B-Instruct**

System prompt:

```
You are Qwen, created by Alibaba Cloud. You are a helpful assistant.
```

User prompt:

```
You are given a Python function and an assertion containing an input to the
  ↪ function. Complete the assertion with a literal (no unsimplified expressions
  ↪ , no function calls) containing the output when executing the provided code
  ↪ on the given input, even if the function is incorrect or incomplete. Do NOT
  ↪ output any extra information. Execute the program step by step before
  ↪ arriving at an answer, and provide the full assertion with the correct
  ↪ output in [ANSWER] and [/ANSWER] tags, following the examples.

[PYTHON]
def f(s):
    s = s + s
    return 'b' + s + 'a'
assert f('hi') == ??
[/PYTHON]
[THOUGHT]
Let's execute the code step by step:

1. The function f is defined, which takes a single argument s.
2. The function is called with the argument 'hi', so within the function, s is
  ↪ initially 'hi'.
3. Inside the function, s is concatenated with itself, so s becomes 'hihi'.
4. The function then returns a new string that starts with 'b', followed by the
  ↪ value of s (which is now 'hihi'), and ends with 'a'.
5. The return value of the function is therefore 'bhihia'.
[/THOUGHT]
[ANSWER]
assert f('hi') == 'bhihia'
[/ANSWER]

[PYTHON]
def f(nums):
    for i in range(len(nums)):
        if not i % 2:
            nums.append(nums[i] * nums[i + 1])
    return nums
assert f([]) == ??
[/PYTHON]
[THOUGHT]
```

---

**Listing 22  Prompt templates for evaluating SemCoder__1030 and SemCoder-S__1030**

Prompt:

```
Simulate the Execution: You are given a Python function and an assertion
  ↪ containing a function input. Complete the assertion containing the execution
  ↪  output corresponding to the given input in [ANSWER] and [/ANSWER] tags.

def f(nums):
    for i in range(len(nums)):
        if not i % 2:
            nums.append(nums[i] * nums[i + 1])
    return nums
assert f([]) == ???
```

**Listing 23   Prompt templates for evaluating the MagiCoder series**

Prompt:
```
You are an exceptionally intelligent coding assistant that consistently delivers
  ↪ accurate and reliable responses to user instructions.

@@ Instruction
You are given a function and an input. Provide the output of executing the
  ↪ function on the input. Surround the output with <answer> </answer> tags. Do
  ↪ NOT output any extra information.

Example 1:

def f(x):
    return x

<input>
17
</input>
<answer>
17
</answer>

Example 2:

def f(x):
    return x

<input>
'a'
</input>
<answer>
'a'
</answer>

Problem:

{code}

<input>
{inputs}
</input>

@@ Response
<answer>
```

**Listing 24   Prompt templates for StarCoder/CodeLlama/DeepSeek-Coder**

Prompt:
```
Based on the given Python code, which may contain errors, complete the assert
  ↪ statement with the output when executing the code on the given test case. Do
  ↪  NOT output any extra information, even if the function is incorrect or
  ↪ incomplete. Do NOT output a description for the assert.

def f(n):
return n
```

```
assert f(17) == 17

def f(nums):
    for i in range(len(nums)):
        if not i % 2:
            nums.append(nums[i] * nums[i + 1])
    return nums
assert f([]) == ???
```

## C.5 Inference Cost of ET-CoT

Because ET-CoT significantly expands reasoning steps, one might be concerned about the resulting inference cost. Therefore, this subsection evaluates how much the generation length increases when ET-CoT is used. First, we compare the generation lengths of the reasoning model Qwen3-8B and ET-CoT. The results are shown in Table 25.

Table 25: Average generation length (in characters) for Qwen3-8B and ET-CoT.

| Model | CRUXEval | LCB | HackerEarth | Aizu | HumanEval |
|-------|----------|-----|-------------|------|-----------|
| Qwen3-8B | 3904.3 | 5490.3 | 3271.8 | 3033.1 | 3056.7 |
| ET-CoT | 2089.7 | 2460.9 | 1835.5 | 1581.9 | 1673.2 |

Although ET-CoT generates very fine-grained reasoning steps, the overall generation length is roughly half of that of the reasoning model. For example, the generation length of ET-CoT is about 53% of that of the reasoning model on CRUXEval and about 44% on LiveCodeBench. This is not merely a difference in generation length, but reflects a fundamental difference in reasoning mode. Reasoning models often perform multiple rounds of backtracking, revising intermediate reasoning steps during inference. This process results in long reasoning token sequences. In contrast, ET-CoT performs a single forward reasoning pass that follows the execution order of the program. As a result, although each step of ET-CoT is more fine-grained, the total generation length tends to be shorter. Also, we believe this reasoning mode is more aligned with the problems, as code execution reasoning is inherently deterministic.

Next, we compare the generation length of ET-CoT with that of a non-reasoning model. Table 26 reports the average token lengths on CRUXEval and LiveCodeBench. For differences among the trace formats, please refer to Subsection 5.4.1.

Table 26: Average generation length (in tokens) for different ET-CoT variants and Llama3.1-8B-Instruct.

| Variant | CRUXEval | LCB |
|---------|----------|-----|
| ET-CoT (Original) | 859.51 | 1698.21 |
| ET-CoT (Removing Loop Internals) | 649.65 | 1297.56 |
| ET-CoT (Minimal Trace) | 642.69 | 1209.15 |
| Llama3.1-8B-Instruct | 209.59 | 249.92 |

Compared with Llama3.1-8B-Instruct, ET-CoT (Original) increases the generation length. The mean output length becomes approximately $4.1\times$ larger on CRUXEval and $6.8\times$ larger on LiveCodeBench. This increase occurs because ET-CoT explicitly generates execution traces, which require additional tokens.

Overall, while ET-CoT increases generation length compared with non-reasoning models, it performs inference with shorter outputs when compared with reasoning models.

## C.6 Additional ET-CoT Fine-Tuning on Qwen2.5-Coder

In the main text, we primarily applied ET-CoT to instruction-tuned models. Here, we additionally report the results of applying ET-CoT to Qwen2.5-Coder-7B-Instruct, which has already been fine-tuned to improve coding ability and is superior to Qwen2.5-7B-Instruct in the benchmarks. The results are shown in Table 27.

Table 27: Effect of ET-CoT on Qwen2.5 and Qwen2.5-Coder models. Best results in each column are shown in bold.

| Model | CRUXEval | LCB | HackerEarth | Aizu | HumanEval | Average |
|---|---|---|---|---|---|---|
| Qwen2.5-7B | 42.38 | 52.82 | 64.61 | 60.18 | 66.10 | 57.2 |
| Qwen2.5-7B + ET-CoT | 70.00 | **88.30** | **77.53** | 76.11 | 71.61 | **76.7** |
| Qwen2.5-Coder-7B | 65.50 | 64.92 | 75.28 | 69.03 | 77.68 | 70.5 |
| Qwen2.5-Coder-7B + ET-CoT | **70.75** | 71.40 | 74.72 | **76.55** | **81.21** | 74.9 |

According to the table, applying ET-CoT to Qwen2.5-Coder-7B-Instruct leads to consistent improvements compared to the original model. When compared with Qwen2.5-7B-Instruct+ET-CoT, the resulting model outperforms it on several benchmarks. However, overall it does not consistently surpass Qwen2.5-7B-Instruct + ET-CoT.

This suggests that stronger natural language-based code reasoning ability does not necessarily lead to better performance after applying ET-CoT. We believe this is because the reasoning mode introduced by ET-CoT is fundamentally different from natural language–based reasoning. When attempting to learn abilities with different characteristics, competition between them is commonly observed (Yu et al., 2020; Springer et al., 2025).

In summary, ET-CoT may not directly build on a model's natural language–based reasoning ability. Instead, it can be understood as a method for introducing a new reasoning mode based on deterministic and fine-grained reasoning steps. This interpretation is also consistent with the results on reasoning models reported in the main text.

### C.7  Match rate between ET-CoT and PyTracify

Table 28: Trace-matching statistics for Llama3.1-8B-Instruct fine-tuned with ET-CoT. We report (i) the proportion of outputs whose execution traces exactly matched those of PyTracify, (ii) the breakdown of correct and incorrect predictions within the trace-matched subset, and (iii) the model's pass@1 accuracy on CRUXEval-O.

| Statistic | Percentage |
|---|---|
| Exact Trace Match Rate | 52.63 |
| Correct among Trace-Matched | 52.38 |
| Incorrect among Trace-Matched | 0.25 |
| CRUXEval-O pass@1 (for reference) | 67.75 |

To assess how much the output traces generated by ET-CoT–fine-tuned models match the ground-truth execution traces produced by PyTracify, we examined the generations of Llama3.1-8B-Instruct fine-tuned with ET-CoT on CRUXEval as an example. According to Table 28, 52.63% of the outputs produced execution traces that matched the ground-truth PyTracify traces exactly. Among these matched cases, 52.38% were correct predictions, while only 0.25% were incorrect despite the traces aligning perfectly. The two mismatched instances correspond to (i) a case where the correct answer consisted of seven consecutive spaces but the model produced an empty string, and (ii) a case where the expected answer was ' 4 2 ' (with leading and trailing spaces), but the model predicted '4 2' without those spaces.

The fact that more than half of the generations reproduced PyTracify's long execution traces exactly provides strong evidence that ET-CoT enables LMs to function as strong code interpreters. Also, even when the execution trace did not match the ground truth, approximately 32% of the predictions were still correct $((67.75 - 52.63)/(100 - 52.63) \approx 32\%)$, indicating that the model fine-tuned with ET-CoT is reasonably robust to errors in its generated traces.

# D    Details of the experiments in Section 5.3

## D.1    Design and preparation of example codes

**Selection of the algorithms.**    We describe how we selected the four iterative algorithms used to evaluate how LMs can repeatedly execute simple operations. In selecting the algorithms, we considered the following factors. (i) Since our goal is to measure the iterative code simulation capability of 8B-class LMs, each step should be sufficiently simple to be solvable even by such models. (ii) We aimed to gain a more fine-grained understanding of LM behavior by tracking the correctness of not only the final outputs but also the intermediate reasoning steps. To ensure that the step count is meaningful, the complexity of each step should be approximately uniform. These correspond to conditions (i) and (ii) in Section 5.3.

With these considerations in mind, we first adopted addition, which has been widely used as a case study for LLMs (Zhong et al., 2023; Yu & Ananiadou, 2024). Next, from representative categories of algorithms, we decided to include one example each from dynamic programming, greedy algorithms, and graph algorithms. After implementing several candidate algorithms from each class and prioritizing conciseness, we ultimately selected jump game DP, interval scheduling of sorted jobs, and BFS, with one representative chosen from each category. Detailed explanations of these algorithms are provided below.

We note that several studies have evaluated the accuracy of LMs on repeated operations under varying levels of complexity (La Malfa et al., 2024; Shojaee et al., 2025). However, these works often employ algorithms with superlinear input complexity to elicit failures in commercial LMs. In contrast, our focus is to examine how the behavior of 8B-class LMs changes before and after ET-CoT training. Accordingly, we restrict our evaluation to relatively simple algorithms whose complexity is linear in the input size.

**Extraction and simplification of iterative parts.**    Because our aim was to measure the step-wise error rate, we eliminated preprocessing and postprocessing to ensure that the difficulty of each step is approximately uniform. For instance, in interval scheduling, we provided a list of jobs that was already sorted by finishing time to satisfy for this reason.

We also took into account the difficulty of each operation for LMs and adjusted the implementation and input/output formats to maximize their likelihood of success at each step. For instance, as in Section 4.2, where we introduced additional training on string manipulations, LLMs generally struggle with string operations. Therefore, in the implementation of addition, we pass digits as lists rather than as plain numbers. Similarly, because LMs often struggle with extracting elements from specified positions in a list, in the addition task we remove the used element from the list at each digit step. Further details are provided in the respective algorithm paragraphs.

**(a) Digit-wise addition (Listing 29).**    Digit-wise addition takes two numbers as input and computes their sum. In implementing this task, we considered the following points.

- LMs generally struggle to extract a specific digit from a number, which causes the difficulty to increase as the computation proceeds step by step. To address this, we decompose the numbers into lists of digits and remove each digit once it has been processed, thereby constructing an equivalent algorithm that is easier for LMs to handle.

- Although `a = A[-1]` and `A = A[:-1]` can be written more compactly as `a = A.pop()`, we keep them separated for clarity. We also avoided to use `.append` and used `[c] + C` instead.

- At every step we print the intermediate result as well as the lists for original numbers `A` and `B`, which ensures that the necessary information remains close to the output even as the computation progresses further away from the input.

- To ensure that the function body consists only of a `while` loop (except for the last two lines), we include not only the input numbers $A$ and $B$ to be added, but also the output $C$ and the carry used in the algorithm as part of the inputs.

The inputs are generated by randomly sampling a pair of $n$-digit numbers, where $n$ is specified as a complexity of the problem. We count one step for each call to print. For completeness of the algorithm, we include an exception handler for the final carry immediately before the output. However, even if a carry occurs in the last digit, we do not count it as a failure. In other words, in $n$-digit plus $n$-digit addition, we evaluate steps only up to $n$, since not every case produces a final carry and including it would make the calculation of the maximum failure rate complicated.

**Listing 29  Python program for (a) digit-wise addition**

```python
def add_equal_length_numbers(A: list[int], B: list[int], C: list[int], carry: int
    ) -> list[int]:
    while A and B:
        a = A[-1]
        b = B[-1]
        c = a + b + carry
        carry = c // 10
        c = c % 10
        A = A[:-1]
        B = B[:-1]
        C = [c] + C
        print(A, B, C, carry)
    if carry:
        C = [carry] + C
    return C
```

**(b) Dynamic Programming for the jump game (Listing 30).** The jump game is described as follows. Given a list of integers `heights`, starting from the leftmost element, one repeatedly "jumps" either one or two positions forward until reaching the right end. At each step, the cost is defined as the absolute difference between the numbers, and the goal is to find a path that minimizes the total cost. This problem can be solved using a standard one-dimensional dynamic programming, where we iteratively update the minimum cost to each position from the left in `DP`.

In implementing this task, we considered the following points.

- To avoid exception handling around the start and end positions, we pre-fill the DP list at positions 0 and 1. This eliminates the need for special cases within the function and ensures that the workload at each step remains uniform. Therefore, a `heights` list of length $n + 2$ leads to $n$ steps of the while loop.

- To eliminate the need for the LM to retrieve values from the middle of the sequence, we remove each element from the `heights` list once it is accessed, since the element will no longer be needed thereafter.

The inputs are generated by randomly sampling from $\{0, \ldots, 9\}^{n+2}$, where $n$ is specified as a complexity of the problem. This function receives `heights` as well as a list `DP`, and fills and outputs this `DP`. We count one step for each call to the print statement.

**Listing 30  Python program for (b) dynamic programming for the jump game**

```python
def jump_DP_easy(heights: list[int], DP: list[int]) -> list[int]:
    while len(heights) >= 3:
        one_step = DP[-1] + abs(heights[2] - heights[1])
        two_steps = DP[-2] + abs(heights[2] - heights[0])
        d = min(one_step, two_steps)
        DP = DP + [d]
        heights = heights[1:]
        print(heights, DP, one_step, two_steps, d)
    return DP
```

**(c) Interval scheduling of sorted jobs (Listing 31).** Given a list of intervals (jobs) specified by `start` and `end` times, the task is to find the maximum number of non-overlapping intervals. This can be solved by first sorting the intervals by their finishing times and then greedily selecting them in order of earliest finishing time. However, since sorting requires $O(n \log n)$ time, it is not suitable for our purpose of evaluating LMs with linear-time algorithms. Therefore, we input a list of jobs which is sorted in increasing order of end time, so that we can isolate only the selection step and use it as the subject of code execution. The output is a boolean list, indicating whether each interval is selected or not.

In implementing this task, as before, we remove each job once it is examined so that the LM does not need to access the middle of the list.

Based on the specified complexity $n$, we generate $n$ jobs. For each job, the start time is sampled from $[0, 2n - 4]$, and the duration (end time minus start time) is sampled from $[1, 4]$. After obtaining the list of intervals, we sort it based on the end time. `last_end` is initialized as $-1$, and `is_selected` is initialized as an empty list. We count one step for each call to print.

**Listing 31    Python program for (c) interval scheduling of sorted jobs**

```python
def interval_scheduling_of_sorted_jobs(jobs: list[tuple[int, int]], last_end: int
    , is_selected: list[bool]) -> list[bool]:
    while jobs:
        start, end = jobs[0]
        jobs = jobs[1:]
        if start >= last_end:
            last_end = end
            is_selected = is_selected + [True]
        else:
            is_selected = is_selected + [False]
        print(jobs, is_selected)
    return is_selected
```

**(d) Breadth First Search (Listing 32).** This task is to perform a breadth-first search from a specified vertex in a connected unweighted graph. In implementing this task, we considered the following points.

- Instead of representing the graph as a list of edges, we construct an adjacency list in advance and provide it to the algorithm as a dictionary. This removes the need for the LM to search for edges at each step.

- The output is defined as the distance from `start_node` to each vertex, which is stored in the form of a dict so that correspondence between each vertex and distance is clear. All values of this dict `Distance` are initialized as $-1$.

- To ensure that the function body consists only of a `while` loop, we initialize the queue outside the function. Specifically, we set `Queue = [start_node]`.

For a given complexity parameter $n$, the input is an undirected graph with $n + 1$ vertices and $2(n + 1)$ edges. We first uniformly sample a minimum spanning tree and then add random edges, thereby ensuring that the graph is connected.

This function takes `Graph`, `Distance`, and `Queue` as inputs, fills `Distance`, and returns it. We define one step as the process from discovering one vertex to discovering the next. Since the `start_node` is already discovered, the total number of steps is one less than the number of vertices. This is the reason why, for complexity $n$, we consider a graph with $n + 1$ vertices.

**Listing 32    Python program for (d) breadth first search**

```python
def breadth_first_search(Graph: dict[int, list[int]], Distance: dict[int, int],
    Queue: list[int]) -> dict[int, int]:
```

```
    while Queue:
        current_node = Queue[0]
        Queue = Queue[1:]
        print(Queue, current_node)
        for neighbor in Graph[current_node]:
            if Distance[neighbor] == -1:
                Distance[neighbor] = Distance[current_node] + 1
                Queue = Queue + [neighbor]
            print(Distance, Queue, current_node, neighbor)
    return Distance
```

## D.2   Evaluation pipeline

We show the prompts used in our experiments in Listing 33. SemCoder (Ding et al., 2024a) and the ET-CoT models were each evaluated using their respective prompt templates (see Subsection C.4). Given that SemCoder is not trained to explicitly output the results of print statements, we evaluated all models based on their final outputs when determining at which step the first error occurred. Concretely, since all programs return either a list or a dictionary, any incorrect element is taken to indicate a failure at the step at which that element should have been produced. In Listing 34, we provide an example of the generated output from Llama3.1-8B-Instruct.

### Listing 33   Sample prompt

System prompt:
```
You are an assistant capable of simulating the execution of Python code. The user
  ↪  will provide Python code that has print statement(s), and your task is to:

1. First, reason through the behavior of the code step by step to understand how
  ↪ the internal variables are updated. Enclose this reasoning between <reason>
  ↪ and </reason> tags.

2. Then, show the complete simulated output of the code, which consists of all
  ↪ printed lines and the returned value. Between <trace> and </trace> tags,
  ↪ write the output from each print call in the order they are executed,
  ↪ placing each one on its own line. Between <answer> and </answer> tags, write
  ↪  the returned value separately, as it would appear in the terminal after
  ↪ running the code.

Use the following format:

<reason>

[your reasoning here]</reason><trace>

[print output here]</trace><answer>

[only the returned value here]</answer>
```
User prompt:
```
def add_equal_length_numbers(A: list[int], B: list[int], C: list[int], carry: int
  ↪ ) -> list[int]:
    while A and B:
        a = A[-1]
        b = B[-1]
        c = a + b + carry
        carry = c // 10
        c = c % 10
        A = A[:-1]
        B = B[:-1]
        C = [c] + C
        print(A, B, C, carry)
    if carry:
        C = [carry] + C
```

```
    return C

A = [2, 1, 5]
B = [8, 6, 0]
C = []
carry = 0

add_equal_length_numbers(A, B, C, carry)
```

**Listing 34  Example of generated outputs (Llama3.1-8B-Instruct)**

```
<reason>

The function `add_equal_length_numbers` performs addition of two numbers
  ↪ represented as lists of digits, using a carry variable to handle cases where
  ↪  the sum of two digits is greater than 9. Here's a step-by-step breakdown of
  ↪  how it works:

1. The function takes four parameters: `A` and `B` which are the lists of digits
  ↪ to be added, `C` which is the list to store the result, and `carry` which is
  ↪  the carry from the previous addition step.
2. The function enters a while loop that continues as long as both `A` and `B`
  ↪ are non-empty.
3. In each iteration of the loop, the last elements of `A` and `B` are extracted
  ↪ using `A[-1]` and `B[-1]`. These elements are added together along with the
  ↪ current `carry` to get the sum `c`.
4. The carry for the next addition step is calculated as `c // 10`. This is done
  ↪ by performing integer division of `c` by 10, which gives the quotient and
  ↪ keeps the remainder.
5. The last digit of the sum `c` is calculated as `c % 10`, which gives the
  ↪ remainder of dividing `c` by 10. This is the new digit to be added to the
  ↪ result list.
6. The last elements of `A` and `B` are removed using `A = A[:-1]` and `B = B
  ↪ [:-1]`.
7. The new digit is appended to the result list `C` using `C = [c] + C`.
8. If there is a carry after the addition, it is set as the new carry for the
  ↪ next step.

The function continues this process until either `A` or `B` is empty. If there is
  ↪  a carry left after the last addition, it is used to add a new digit to the
  ↪ result.

Finally, the function returns the result list `C` after all additions have been
  ↪ performed.

</reason><trace>
[2, 1, 5] [8, 6, 0] [5, 7, 5] 0
[1, 5] [6, 0] [7, 5, 5] 0
[5] [0] [5, 5, 5] 0
[] [] [5, 5, 5] 0
[] [] [5, 5, 5]
</trace><answer>
[5, 5, 5]
</answer>
```

### D.3 Full results

Here we provide the full results for Figures 6 and 7. Figure 10 corresponds to Figure 6, and Figure 11 corresponds to Figure 7, respectively.

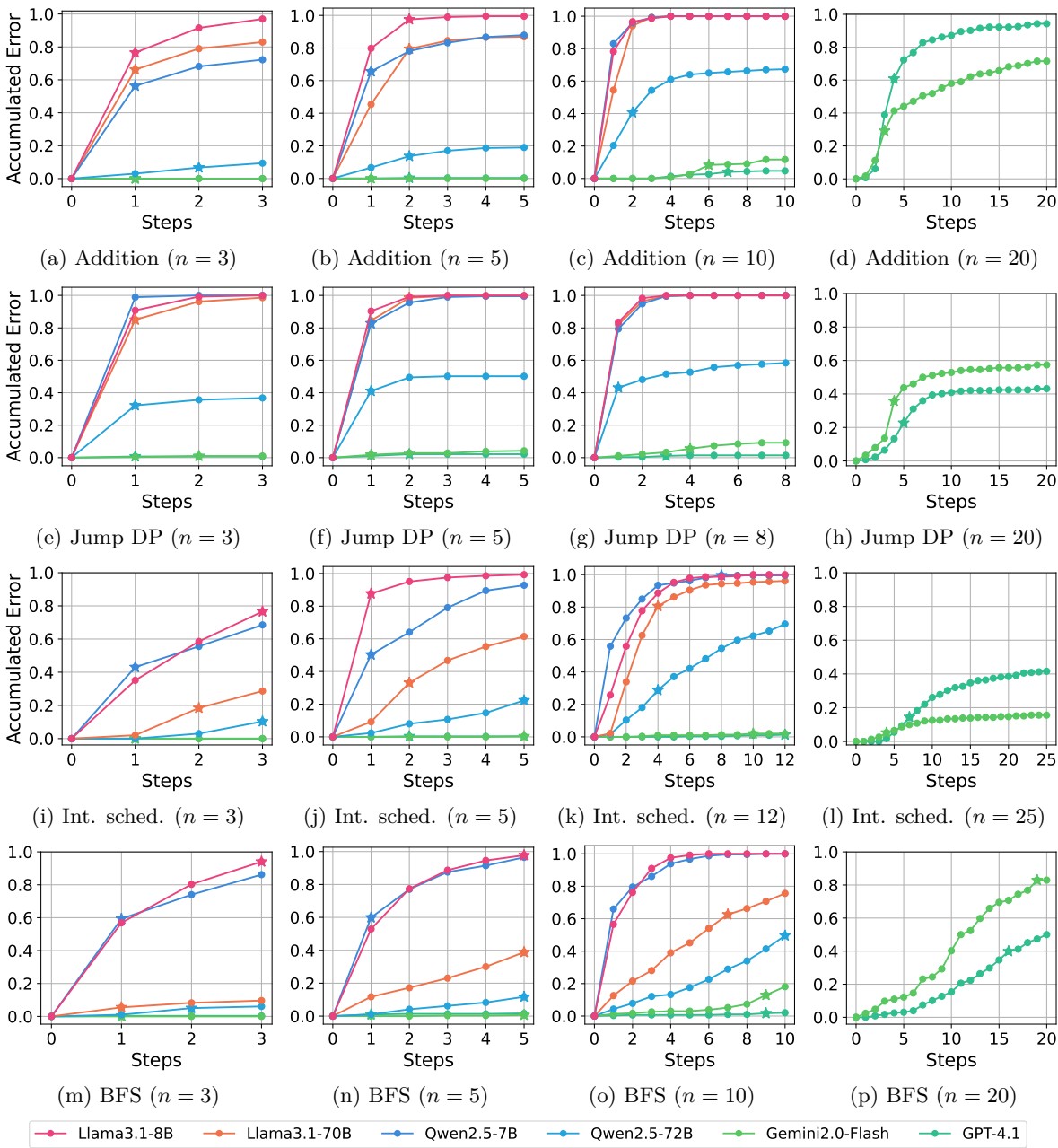

Figure 10: Step-wise cumulative error rate for iterative operations (before ET-CoT, full results).

We provide an overview of the results in Figure 10. First, Llama3.1-8B and Qwen2.5-7B fail on all tasks for $n = 3, 5$ with high probability. In contrast, for Llama3.1-70B, Qwen2.5-72B, Gemini2.0-Flash, and GPT-4.1, the following tendency can be observed, especially for $n \geq 5$. In (d) BFS, the cumulative error rate increases gradually, whereas in (a) addition, (b) jump game DP, and (c) interval scheduling, the step-wise failure rate is higher in the early steps. One hypothesis for this task-specific difference is that BFS is graph-based, unlike addition, jump game DP, and interval scheduling. We note that we obtained partial results for minimum

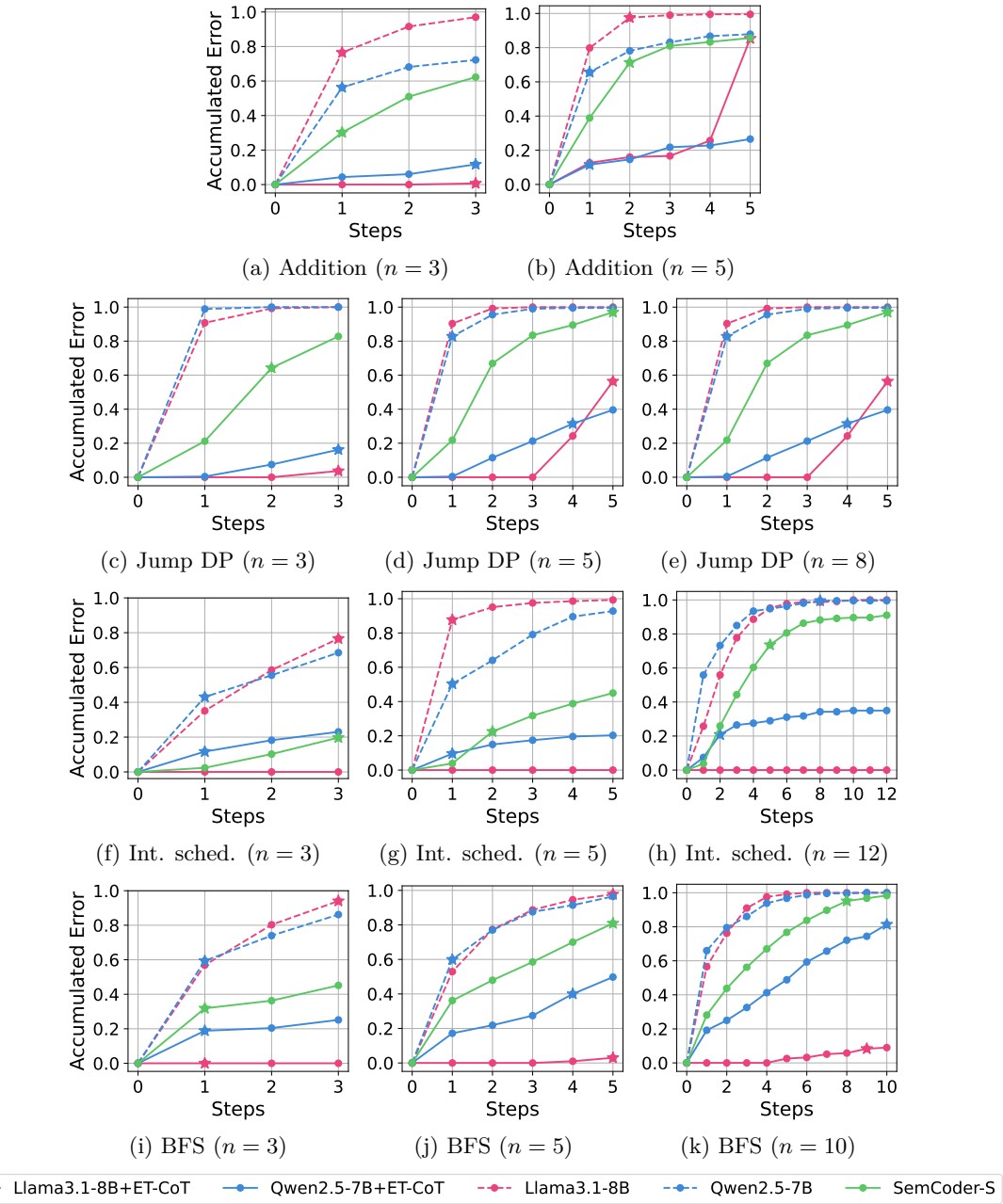

Figure 11: Step-wise cumulative error rate for iterative operations (after ET-CoT, full results).

spanning tree. Because we ultimately adopted BFS, we did not collect a complete set of experimental results for the minimum spanning tree. However, the partial results we obtained exhibited a similar pattern to BFS, with the error rate increasing gradually. Nevertheless, this contrast between the two modes is intriguing, and rather than drawing general conclusions from only five algorithms, further dedicated exploration would be more valuable.

Finally, regarding Figure 11, the improvements from ET-CoT over the base models are substantial across all tasks and all complexities for both Llama3.1-8B and Qwen2.5-7B, and they are also more stable than SemCoder. In particular, Llama3.1-8B+ET-CoT completely suppresses the initial instability.

# E    Example of CER tasks that inherently require long chain of thought

ET-CoT is grounded in the view that code execution tasks can be solved deterministically by stacking fine-grained reasoning steps as a chain of thought. One might argue that, if a model can be trained to generate the answer directly, such an approach would be more efficient. However, we point out that this strategy faces fundamental theoretical limitations, and that concrete examples exhibiting these limitations are relatively simple problems.

Specifically, L and NL denote the classes of problems solvable by deterministic and nondeterministic Turing machines, respectively, using logarithmic space in the input size $n$. Roughly speaking, NL is the class decidable in logarithmic space nondeterministically. NL-hard denotes the set of problems to which every problem in NL reduces within log space. L is the class of problems decidable deterministically in logarithmic space, simpler than NL. By borrowing Merrill & Sabharwal (2024), the following statement can be made.

**Proposition 1.** *Assuming* L $\neq$ NL*, solving an* NL*-hard problem requires a number of CoT steps that scales with the input size $n$. More concretely, solving an* NL*-hard problem with a log-precision ($O(logm)$ precision for $m$ decoding steps) decoder-only transformer with strict causal masking (each position attends only to earlier tokens), saturated attention (idealized hard attention), and projected pre-norm (apply linear projection before layer normalization for each sublayer) requires $\omega(\log n)$ intermediate decoding steps as a function of the input length $n$.*

To confirm this, suppose that there exists an NL-hard problem $A$ that can be solved by such a Transformer with $O(\log n)$ intermediate steps to derive contradiction to Theorem 4 of Merrill & Sabharwal (2024).

A canonical NL-complete problem (one of the hardest problems in the class NL) is reachability in directed graphs (Sipser, 1996). Its algorithm can be written in roughly ten lines, and we also considered it in Section 5.3 (putting aside our algorithm's linear memory usage). Thus, the fact that even this familiar example exhibits limitations of direct-output approaches shows the need for problem-adaptive, long chain of thought as employed in ET-CoT.

