# OpenReview forum: "Transforming Language Models into Program Interpreters via Execution Trace Chain of Thought"
_TMLR — Accepted by TMLR_

### Review · Reviewer_3hzs · 2026-02-14

**Summary Of Contributions:**

This paper presents a fine-tuning approach to improve LLM performance on code execution reasoning (CER) tasks, where a model must predict the output of a code snippet without actually executing it. The authors analyze failures of small LLMs (e.g., ~8B models) on CER, arguing that small models is weak learning basic syntax. They propose ET-CoT, which fine-tunes models to generate sub-line, execution-style traces generated by the proposed PyTracify method. PyTracify traces the execution and generates stringified version of Python AST, which guide models better learning code syntax interpretation. Experiments report large gains for 8B models across CER benchmarks.

Strength:
* The paper is well-motivated by the limitation of understanding code syntax for small models.
* Fine tuning on intermediate steps for coding (e.g traces, compiler steps) has been proven effective research. The proposed method can be seen as an effective method to augment the training data.
* Experiment results show large improvement for the proposed method over baseline and other fine-tuning method such as SFT.

Weakness:
* In sec 3, the paper claimed "Most 8B-class models have limitations in basic syntax understanding even with CoT: except for Qwen3-8B". However, models like Llama 3.1 or Mistral are not trained or fine-tuned with CoT data. Adding "reason" section in the system prompt is usually not very effective for such models to generate valid reasoning steps. While Qwen3-8B, which is a more recent models and has been trained on CoT data, as mentioned by the author is an exception, I think this data point is important, rather than a simple exception. It suggests that small models can learn coding capability well as long as trained from CoT data (or distilled from larger models such as larger models in Qwen3 family)
* While several experiment results show large improvement over baseline, non reasoning models like llama or mistral are very weak baseline. For such models, there are many methods for fine tuning to improve the performance on specific tasks. The paper explored SFT or GRPO, but doesn't consider fine tuning method that involves CoT. SOTA small models usually leverage frontier models to generate CoT for training, given it can better uses learning budget to learn on tokens that are mostly effective. The data on QWen-3 is weaker, it is also not clear that when models are sufficiently post-trained on CoT data, the proposed method can further improve the performance.

**Audience:**

Yes

**Audience Explanation:**

Coding is important capability for LLM and has been getting great popularity in research.

**Broader Impact Concerns:**

N/A. This paper is about coding capabilities for LLM.

**Claims And Evidence:**

No

**Claims Explanation:**

See weakness above. Current evaluation is dominated by non-reasoning baselines, so it’s unclear whether the proposed method still yields meaningful gains once a model already has strong inherent CoT ability.

**Requested Changes:**

* Add results on more recent reasoning-capable small models (e.g OLMo 3 thinking models) (results on QWen3-8B are interesting, but more evidence is needed).
* Consider adding discussion around today’s small-model landscape: many recent 7–10B models are post-trained/distilled for reasoning. The paper should explain whether ET-CoT is a substitute for standard CoT distillation, or be complementary.

---

### Review · Reviewer_hudS · 2026-02-16

**Summary Of Contributions:**

The paper addresses the limitations of 8B-class Language Models (LMs) in Code Execution Reasoning (CER) by proposing ET-CoT (Execution Trace Chain of Thought), a fine-tuning method utilizing extremely granular execution traces generated by a custom interpreter, PyTracify. Unlike previous methods that log variables per line, PyTracify expands expressions at a sub-line level (e.g., resolving binary operations and comparisons explicitly).

Key contributions include:
1. Developing a custom interpreter, PyTracify, and a dataset of 127k examples including custom subsets for string manipulation and token length to address specific LM weaknesses.
2. Fine-tuning 8B models (Llama 3.1, Qwen 2.5) with ET-CoT, yielding significant gains on benchmarks like CRUXEval and LiveCodeBench and often outperforming domain-specific "Coder" models.
3. Identifying that LMs struggle with 'initial instability' in iterative tasks and demonstrate that the ability to explain code in natural language does not correlate with the ability to execute it.
4. Conducting an ablation study to confirm that trace granularity is critical for performance.

**Additional Comments:**

Overall, the paper is well-structured and the distinction drawn between natural language explanation and procedural execution is an insightful finding for the field.

**Audience:**

Yes

**Audience Explanation:**

This paper is relevant to researchers focused on enhancing the reasoning capabilities of smaller language models. The findings regarding the dissociation between natural language explanation capabilities and actual code execution fidelity provide insight into the limitations of current interpretability methods.

**Broader Impact Concerns:**

The authors acknowledge limitations regarding the integration of CER with code generation, and the paper primarily focuses on improving the correctness of code execution simulation, which generally aids in debugging and safety verification rather than introducing new risks of generating harmful content. Standard considerations regarding LLMs apply but are not unique to this specific method.

**Claims And Evidence:**

Yes

**Claims Explanation:**

The authors provide sufficient evidence to support their claims:
1.  The performance improvements are demonstrated across multiple model families and benchmarks like CRUXEval, LiveCodeBench, HumanEval, detailed in Table 2, which shows consistent performance gains over base models.
2. Improvements are verified against baselines including direct fine-tuning (SFT), reinforcement learning (GRPO), and prior methods like SemCoder, as illustrated in Figure 1(b) and Table 2.
3. Ablation studies in Section 5.4 quantify the impact of trace granularity, showing that full traces yield higher accuracy than coarser formats (Table 4). Additionally, the necessity of the custom dataset is supported by Table 5, which demonstrates performance degradation when specific subsets are removed.
4. Finally, the analysis of initial instability in iterative operations are visualized in Figures 6 (before ET-CoT) and Figure 7 (after ET-CoT) providing quantitative evidence for the method's efficacy in stabilizing reasoning.

**Requested Changes:**

I recommend the following additions to strengthen the work:
1. While the authors mention that traces are long and briefly note computational costs for training, a more explicit discussion on the inference latency/cost for the user would be beneficial. ET-CoT significantly expands the token output (e.g., mean token length ~923 ). A brief comparison of inference time vs. standard execution or direct prediction would help practitioners weigh the trade-offs.
2. Section 5.2 notes that ET-CoT slightly degraded the performance of Qwen3-8B, due to the model's 'inherently reasoning-oriented nature'. Analyzing the interference between the model's internal pre-trained reasoning patterns and the imposed PyTracify format to understand _why_ such a conflict occurs would provide valuable insights for expanding this work to future reasoning-heavy models.

---

### Review · Reviewer_xCbP · 2026-02-17

**Summary Of Contributions:**

The paper presents ET-CoT, a new approach for improving LLM's ability to act as program interpreters and execute code. The approach consists of fine-tuning a model on the task of code execution using detailed execution traces as CoT rationales. Specifically, the paper introduces PyTracify, a specialized Python interpreter that produces textual traces that can be used as CoT rationales for fine-tuning the models. Experiments on five code execution benchmarks shows the proposed ET-CoT can significantly improve the performance of small 7/8B general LLMs.

**Audience:**

Yes

**Audience Explanation:**

Yes, the topic of LLMs for code-related tasks has received significant attention in the ML community and is likely to be of interest to significant portion of TMLR's audience.

**Broader Impact Concerns:**

I have no concerns.

**Claims And Evidence:**

No

**Claims Explanation:**

1. *Motivation*: the paper considers a very specific goal which I think is not sufficiently motivated. While improving LLM's code reasoning and interpretation skills is an important goal, it is not clear:
    - whether this skill is important on its own, i.e., a model that only does that, rather than part of a the skill-set of modern coding agent? (is there a need for such capability on its own?)
    - why is it important to have small models (specifically, 8B) capable of performing this task?


2. *Experiments*: I have concerns regarding the experimental setting and analysis which are not sufficient to justify the gains of the proposed methodology.
    - It is not clear why the paper focuses exclusively on 7/8B models alone. It is unclear if slightly larger models (not very large, but for example 16B/32B) would still exhibit similar gains. Further, I imagine that the focus exclusively on 8B models has led to the exclusion of some strong coding models like Qwen3 coder (whose smaller model is 30B but with only ~3 active ones). I think the paper would benefit from experiments with at least one more size (e.g., ~30B) to establish the gains hold for these models as well.
    - It is not clear why Qwen2.5-Coder-7B-Inst (that performs quite well) has not been tested with ET-CoT. It would also be interesting to see if it improves Qwen3 coder model that was not included (perhaps because it is bigger than 8B - see my comment above).
    - The gain for Qwen3-8B is very limited. Given that this is a general purpose and very small model, the slight improvement due to ET-CoT fine-tuning for one specific task is not too surprising. Further, it seems the the fine-tuning has hurt the basic syntax understanding of Qwen 3. This results strengthen the concern that slightly bigger/stronger models may not benefit from the proposed approach.
    - I don't think it is a critical flaw but experiments focus on one programming language. It is not clear whether this approach extends to additional languages and whether similar gains would be observed for these languages. I understand the available benchmarks may be limited to Python.

**Requested Changes:**

- Provide motivation for the narrow current setting (small and specialized model for the task of code program interpretation) [critical]
- Show results for Qwen2.5-Coder-7B-Inst with ET-CoT [critical]
- Experiments on at least one more model size to see gains are not restricted to the (relatively small) size of 8B. [important]
- Experiments with one more programming language [optional]

---

### Decision · Action_Editor_PAmo · 2026-04-30

**Recommendation:** Accept with minor revision

**Additional Comments:**

The authors have largely already updated the manuscript -- however, the current version highlights the changes for the benefit of the reviewers. The authors should incorporate these as part of the main text and make any remaining minor changes that may be required.

**Audience:**

Yes

**Audience Explanation:**

The ability of LLMs to perform code execution reasoning is an important aspect of LLM-based coding. It also is related to reasoning abilities more generally and so the topic will be of interest to a sub-community within the wider ML area.

**Claims And Evidence:**

Yes

**Claims Explanation:**

The reviewers are all satisfied that the claims made in the paper are justified properly. The concerns expressed are about insufficient experiments and significance. More experiments have been conducted by the authors after the initial reviews and these have satisfied the reviewers to a large extent.